# Real-Time Temperature Distribution Monitoring in Chinese Solar Greenhouse Using Virtual LAN

**Shiye Yang** [1] , **Xin Liu** [1] , **Shengyan Liu** [1], **Xinyi Chen** [1] and **Yanfei Cao** [1,2,*]

1 College of Horticulture, Northwest A & F University, Yangling 712100, China; ysy0819@nwafu.edu.cn (S.Y.); 2020050377@nwafu.edu.cn (X.L.); liuas@nwafu.edu.cn (S.L.); chenxyiii@nwafu.edu.cn (X.C.)

2 Key Laboratory of Protected Horticulture Engineering in Northwest, Ministry of Agriculture and Rural Affairs, Northwest A & F University, Yangling 712100, China

* Correspondence: caoyanfei@nwsuaf.edu.cn

**Abstract:** The internal air temperature of Chinese solar greenhouse (CSG) has the problem of uneven spatial and temporal distribution. To determine temperature distribution at different locations, we designed a greenhouse temperature real-time monitoring system based on virtual local area network (VLAN) and estimate, including interpolation estimation module, data acquisition, and transmission module. The temperature data were obtained from 24 sensors, and the Ordinary Kriging algorithm estimated the temperature distribution of the whole plane according to the data. The results showed that the real-time temperature distribution monitoring method established was fast and robust. In addition, data validity rate for VLAN technology deployed for data transmission was 2.64% higher than that of cellular network technology. The following results are obtained by interpolation estimation of temperature data using gaussian model. The average relative error (ARE) of estimate, mean absolute error (MAE), root mean square error (RMSE), and determination coefficient ($R^2$) were $-0.12\,°C$, $0.42\,°C$, $0.56\,°C$, and 0.9964, respectively. After simple optimization of the number of sensors, the following conclusions are drawn. When the number of sensors were decreased to 12~16, MAE, RMSE, and $R^2$ were 0.40~0.60 °C, 0.60~0.80 °C, and >0.99, respectively. Furthermore, temperature distribution in the greenhouse varied in the east–west and north–south directions and had strong regularity. The calculation speed of estimate interpolation algorithm was 50~150 ms, and greenhouse Temperature Distribution Real-time Monitoring System (TDRMS) realized simultaneous acquisition, processing, and fast estimate.

**Keywords:** greenhouse; temperature distribution; real-time monitoring; ordinary kriging; LabVIEW

## 1. Introduction

With the rapid development of protected horticulture, the area under cultivation has expanded exponentially in recent years. Statistics indicate that, as of 2018, total area under facility planting in China stood at 1.894 million hectares, including 577 thousand hectares under solar greenhouse agriculture [1]. These data show the growing importance of facility agriculture in off-season supply of vegetables. Presently, the COVID-19 pandemic and the volatile security situation in Russia and Ukraine have exacerbated the global energy crisis. Therefore, for stable production and supply of vegetables, it is particularly important that new agricultural production units such as greenhouses are developed. In CSG, temperature is an important factor affecting crop quality and yield. However, greenhouse environment is influenced by external environment and human factors [2]. Proper temperature regulation is an essential management aspect of greenhouse operations. Winter heating and summer cooling have become an important research direction of greenhouse temperature regulation. In most cases, temperature distribution inside a greenhouse is considered to be uniform in using temperature measurements at the center of the greenhouse or a few characteristic temperatures to represent the overall temperature

of the greenhouse. However, studies have shown that temperature distribution in the greenhouse is uneven due to the effect of external weather conditions, including outdoor solar radiation and temperature and internal conditions such as greenhouse crops, soil, and structural design [3,4]. The temperature has strong temporal and spatial distribution. A high temperature gradient in the greenhouse has a negative impact on greenhouse production [5].

Many studies have considered real-time monitoring of temperature variation and distribution patterns in the greenhouse [6,7], and they use a variety of environmental monitoring sensors to obtain environmental data inside and outside the greenhouse, and use different methods to build models of parameters such as temperature inside the greenhouse. Presently, Computational Fluid Dynamics (CFD) software is widely used to simulate various aspects of greenhouse environment, such as simulating temperature field distribution of the wall to determine the thickness of the wall [8], horizontal and vertical temperature fields [9–12], and humidity field [3,13]. Temperature is unevenly distributed in both horizontal and vertical directions, as is humidity. Using CFD software can accurately reflect the actual distribution of greenhouse temperature and provide a new approach for simulating greenhouse temperature. In addition, many researchers have established highly accurate and more realistic greenhouse temperature prediction models using machine learning. These models accurately reflect changes and laws of greenhouse temperature. For example, LSSVM (Least Squares Support Vector Machines) optimized by improved PSO (Particle Swarm Optimization) [14], modeling greenhouse environment using SVM (Support Vector Machines) [15], convex bidirectional extreme learning machine [16], and different neural network algorithms [5,17–19] are used to predict and simulate environmental factors such as greenhouse temperature. In particular, CFD software has high simulation accuracy and offers additional advantages, including exploration of the distribution law of greenhouse environmental factors. However, the calculation time is considerably long. Therefore, when applied to regulation of greenhouse environment, it may not provide timely information to guide greenhouse producers. Machine learning has a small error rate and good predictability when used to simulate environmental factors such as greenhouse temperature. However, this method mainly simulates and predicts values where the representative location within the greenhouse of one to several key parameters of environmental factors is useful for greenhouse modeling and distribution law of environmental factors. Therefore, for a rapid estimate of temperature field in the greenhouse, interpolation methods are used to predict related greenhouse variables.

Xiao et al. [20] used Cubic, Natural, and Liner interpolation methods to visually estimate the temperature field of the greenhouse. The average error was 1.5 °C and the Cubic interpolation calculation was relatively accurate for simulating temperature field at a specific time in the greenhouse. Bojacá et al. [21] introduced geostatistics into the greenhouse temperature estimate field and used R software to interpolate the horizontal temperature of an experimental greenhouse. The average error between estimated dataset and observed dataset was 0.4 °C. Zhang et al. [22] used simple Kriging geostatistical interpolation to estimate the temperature field of plant canopy in the solar greenhouse and verified the interpolation in sunny, cloudy, and rainy days. The average root mean square errors were 1.34 °C, 0.95 °C, and 0.40 °C for sunny, cloudy, and rainy days, respectively. Geostatistical interpolation is more appropriate for different estimate interpolation analyses of complex greenhouse temperature variables. However, real-time estimate of greenhouse temperature field may not be achieved using geostatistical estimates, and the software does not provide basic real-time temperature distribution data for formulating an effective temperature control strategy.

Therefore, to compensate for the defects of the methods discussed and achieve real-time analysis of the temperature distribution of greenhouses, a new tool is needed for geostatistical modeling. The present study combined virtual local area network (VLAN) [23] technology and geostatistical estimate methods to derive interpolations of the temperature field in the horizontal plane. We also compared the advantages and disadvantages of

cellular network (CN) [24] technology and VLAN technology in transmitting data and analyzed interpolation effects after optimizing the sensors. Aiming at the uneven temperature distribution in CSG, a real-time estimate method of CSG temperature distribution based on VLAN technology was proposed to realize real-time monitoring of CSG temperature distribution and further understand the specific situation of temperature distribution.

## 2. Materials and Methods

### 2.1. Description of Experimental Greenhouse

The experimental greenhouse was located in the horticulture field of Northwest A & F University in Yangling Agricultural High-tech Industrial Demonstration Zone, Shaanxi Province, 107°59′ E~108°08′ E and 34°14′ N~34°20′ N. This location belongs to warm temperate semi-humid and semi-arid climatic zones in East Asia and experiences a typical continental monsoon climate. The experimental greenhouse was a modular assembled solar greenhouse divided into three compartments according to the different types of the northern wall (Earthen wall, water wall, gravel wall), separated by polystyrene board. This experiment was conducted in a compartment bounded by an earthen wall covered externally with cement mortar in the north and a brick wall covered externally with polyurethane panels in the west. The greenhouse was 10.5 m and 15.0 m long in the north–south and an east–west direction, respectively. The number 1 in Figure 1a is the experimental greenhouse. Figure 1b shows the interior and exterior of the earth-walled greenhouse.

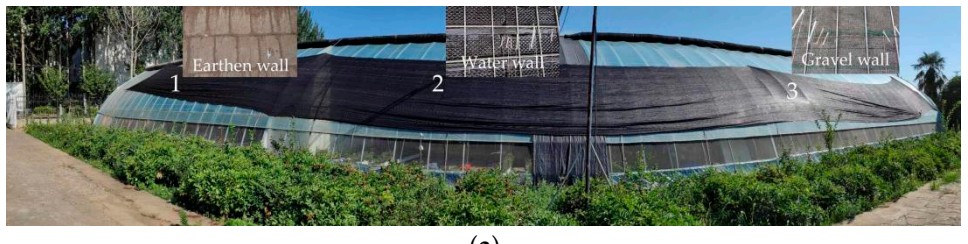

(**a**)

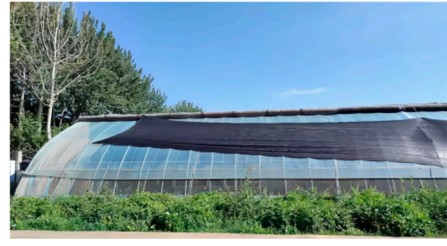 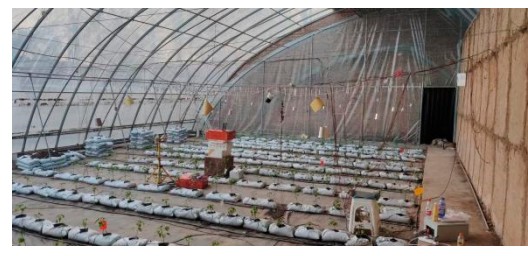

(**b**)

**Figure 1.** CSG real figure. (**a**) is a full-frontal view of the experimental CSG. (**b**) Earthen wall module of experimental greenhouse.

### 2.2. Experimental Design

In the 15.0 m × 10.5 m test area, 28 Pt100 temperature sensors were installed at a height of 1.5 m to measure temperature at different locations in the greenhouse. The solar radiation sensor was located at the center of the greenhouse horizontal plane at 0.8 m to measure the solar radiation, as shown in Figure 2.

### 2.3. Introduction of Real-Time Monitoring System for Temperature Distribution in a CSG

Greenhouse Temperature Distribution Real-time Monitoring System (TDRMS) consisted of two parts: data acquisition part and data estimate part. The data acquisition part included data acquisition module and data storage module. The data estimate part included estimate algorithm module and estimate image output module. TDRMS real-time input included temperature data collected by Data Acquisition Module (DAM). These

data were then passed into the temperature interpolation estimate module for real-time monitoring of temperature distribution. The system architecture is shown in Figure 3.

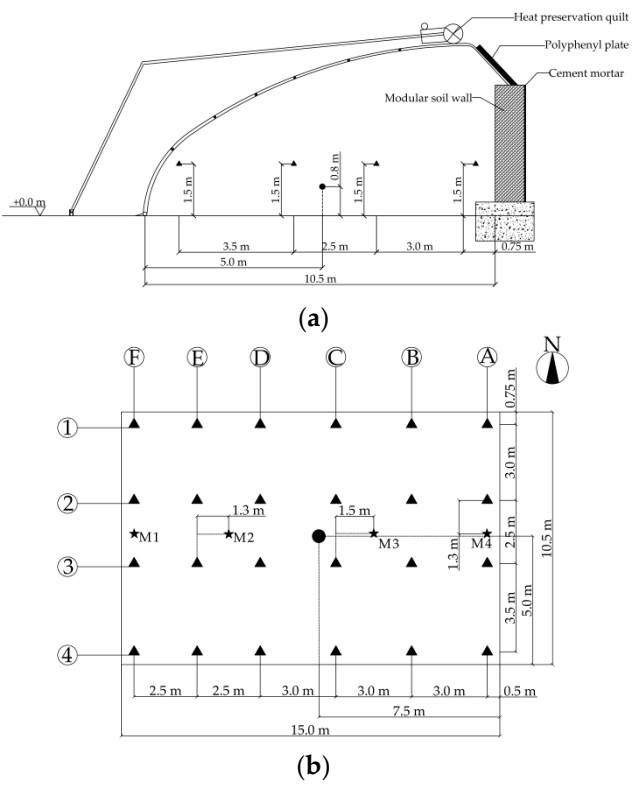

(a)

(b)

**Figure 2.** Distribution of indoor temperature and solar radiation sensors. (**a**) The distribution of temperature and solar radiation sensors in the vertical direction; (**b**) The distribution of temperature and solar radiation sensors on the horizontal plane. ▲ is Pt100 temperature sensor; ● is solar radiation sensor. ★ is Pt100 temperature sensor used to obtain the observation value and analyze the estimate interpolation effect. Where 1 to 4 are row marks; A to F are column marks; M1 to M4 are sensor numbers.

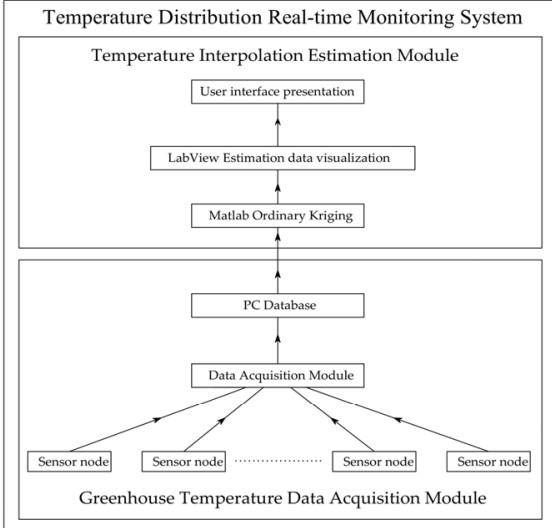

**Figure 3.** Greenhouse Temperature Distribution Real-time Monitoring System (TDRMS) flow diagram.

### 2.3.1. Introduction of Interpolation Estimate Module

The interpolation estimate algorithm is the ordinary Kriging interpolation in geostatistics, and the interpolation algorithm is run by calling the DACE (Design and Analysis of

Computer Experiments) tool library in MATLAB R2021a software (MathWorks Inc., Natick, MA, USA). The DACE tool library automatically adjusts the parameters of the semivariogram [25] to achieve the best estimate effect. In this experiment, the test area where the north wall was earthen was divided into grids of equal rows and columns, and each grid was filled with estimate data. Different data were given different pseudo-colors in TDRMS, and then we used Laboratory Virtual Instrument Engineering Workbench (LabVIEW)'s powerful user interface to visually display temperature data.

LabVIEW used in TDRMS is a graphical programming environment developed by National Instruments, whereas MATLAB is a mathematical software developed for MathWorks. LabVIEW provides a large number of integrated function libraries, which is more useful for designing human–computer interaction interface, supports instrument programming, equipment communication and data acquisition, and has a powerful visualization function. MATLAB, as a mathematical software, offers great advantages in matrix operation processing. Therefore, combining LabVIEW and MATLAB provides a fast real-time interpolation estimate of greenhouse temperature distribution.

### 2.3.2. Data Acquisition and Transmission

Temperature acquisition hardware adopted standard MODBUS TCP (Modicon, MA, USA) protocol for industrial DAM. It was equipped with 32-bit ATMEL ARM high-speed processor acquisition accuracy of 0.1 °C. The temperature data of observed values used DAM (M1 and M4 are shown in Figure 2), including analog quantity of data interface and Pt100 acquisition interface. The accuracy of Pt100 temperature measurement and analog quantity of data were 0.1 °C and 0.1%, respectively. Data transmission equipment included Oraybox-X4C wireless router (Oray, Shanghai, China) and USR-DR154 guide rail type DTU (Data Transfer Unit).

In this experiment, data collected by VLAN and CN were used to compare efficiency and stability between the two methods. When the CN technology of Internet of Things (IoT) [26] platform is used for data collection, the electrical signal of Pt100 is input into the DAM through three-core line, while the DAM is connected to 4G DTU through the RS-485 interface. The Modbus protocol is used for data transmission. Modbus protocol in 4G DTU is converted into Message Queuing Telemetry Transport (MQTT, IBM, NY, USA) protocol. The data are transmitted to the IoT platform through the cellular network technology and downloaded to the personal computer for analysis and use. The architecture is shown in Figure 4a. On the other hand, when TDRMS is used to collect data, Pt100 electrical signal is transmitted to DAM through the three-core wire using TCP transmission protocol. DAM is connected with the router through the RJ45 crystal interface. The router and the personal computer are connected via VLAN technology. TDRMS is used to send Modbus TCP instruction message to DAM. TDRMS receives a response message from DAM and extracts data bytes in the returned message to convert them into decimal temperature data that are stored in a database. The architecture is shown in Figure 4b. Compared with CN technology transmission, VLAN technology can save a one-step protocol conversion process, reducing the possibility of errors in the transmission process.

### 2.4. Temperature Interpolation Estimate Principle

In geostatistical interpolation, ordinary Kriging method is used for temperature data interpolation. Kriging method is based on spatial autocorrelation. It uses raw data and semivariogram to correctly estimate unknown sampling points of regionalized variables [27]. In geostatistics, Kriging is used for interpolation estimate of large-scale geostatistical elements such as precipitation, temperature, and altitude. Unlike spatial deterministic interpolation such as inverse-distance-weighted interpolation, global polynomial interpolation, and local polynomial interpolation, ordinary Kriging interpolation is a spatial non-deterministic interpolation approach. The Kriging interpolation principle formula can be expressed as:

$$Z*(s) = \mu(s) + \varepsilon(s) \tag{1}$$

In the formula, *s* is the point of different positions, namely spatial coordinates; Z*(*s*) is a variable at *s*, which can be broken down into deterministic trend values $\mu(s)$ and autocorrelation random errors $\varepsilon(s)$.

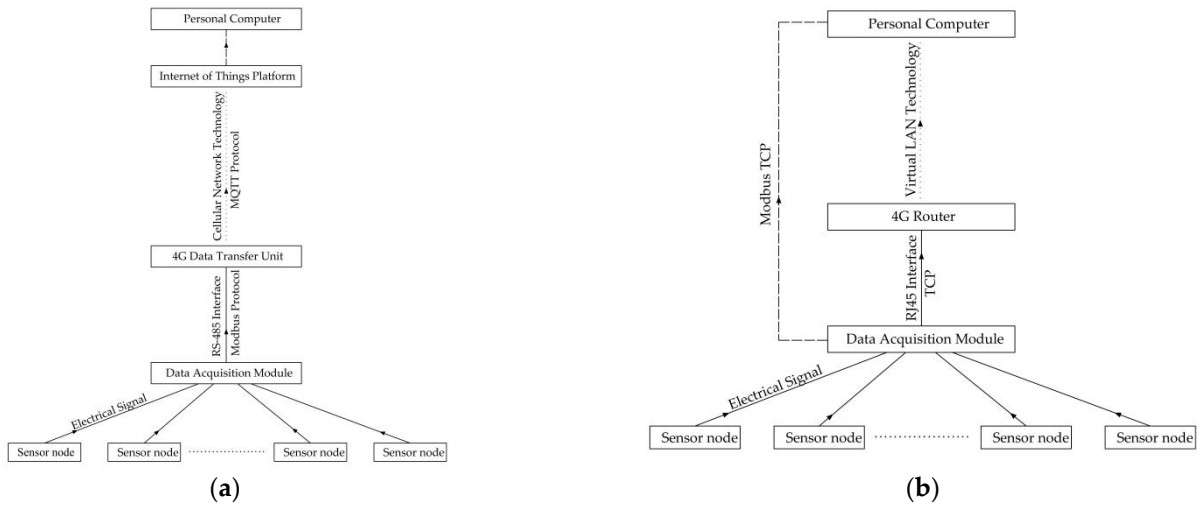

**Figure 4.** The frame structure of two data transmission methods. (**a**) Cellular Network technology data transmission structure; (**b**) Virtual LAN technology data transmission structure.

Kriging interpolation determines the optimal weight of interpolation estimate based on unbiased prediction and minimum variance. Therefore, Kriging interpolation needs to meet the following two conditions:

(1)    Unbiased conditions:

$$E[Z*(s_0) - Z(s_0)] = 0 \tag{2}$$

(2)    Optimal conditions:

$$\min \mathrm{V}ar[Z*(s_0) - Z(s_0)] \tag{3}$$

In the formula, Z*($s_0$) is the estimated value at the known point $s_0$, and $Z(s_0)$ is the observed value at $s_0$. For unbiased condition to be met, the expected value of the difference between observed value of the known point and estimated value must be the same. The optimal condition is one with the smallest variance between observed value of the known point and estimated value.

*2.5. Semivariogram Model Selection*

In the selection of semivariogram, gaussian model and spherical model are mainly used. In DACE, Gaussian model and spherical model are expressed in the following forms:

(1)    Spherical model:

$$1 - 1.5\xi_j + 0.5\xi_j^3, \ \xi_j = \min\{1, \theta_j |d_j|\} \tag{4}$$

(2)    Gauss model:

$$\exp(-\theta_j d_j^2) \tag{5}$$

In the formula, $\xi_j$ is the independent variable at *j*. $d_j$ is a parameter with spatial distance correlation at *j*. $\theta_j$ is the parameter of the relevant model at *j*.

It can be concluded from Table 1 that there is little difference between the two kinds of model in cross-verification, so the model is subsequently verified.

**Table 1.** Cross-validation for precision evaluation.

| | SE | $R^2$ |
|---|---|---|
| Spherical model: | 0.022 | 0.9949 |
| Gauss model: | 0.029 | 0.9931 |

SE is the standard error of regression coefficient, and the smaller the value, the higher the accuracy; $R^2$ is the square of the correlation coefficient, and the larger the value, the better the regression effect.

By comparison, it can be found that the difference estimation effect of zero-order polynomial of Gaussian model is the best, and the results are shown in Table 2. Before kriging interpolation estimation, the temperature data of observation points were analyzed normally, and the data distribution was in accordance with the normal distribution according to the normal Q-Q diagram (see Appendix A Figure A1 for the Q-Q diagram).

**Table 2.** Different model fitting effects.

| | | Decision Coefficient ($R^2$) | Mean Absolute Error (MAE)/°C | RMSE/°C |
|---|---|---|---|---|
| Spherical Model | Zero order polynomial | 0.9957 | 0.51 | 0.65 |
| | First order polynomial | 0.9941 | 0.51 | 0.70 |
| | Second order polynomial | 0.9930 | 0.49 | 0.72 |
| Gaussian Model | Zero order polynomial | 0.9964 | 0.42 | 0.56 |
| | First order polynomial | 0.9944 | 0.46 | 0.65 |
| | Second order polynomial | 0.9929 | 0.50 | 0.72 |

*2.6. Test Method*

The temperature sensor adopted the Pt100 platinum thermal resistance chip of Heraeus, Germany. The signal line adopted three-core tetrafluoro silver plating shielding line. The probe was 4 mm in diameter and 30 mm in length. Its measurement range is $-50\sim200$ °C, with an accuracy of 0.1 °C. The temperature sensor probe was shielded by a reflective shielding device consisting of a disposable paper cup coated with aluminum foil to avoid direct exposure to the sun. Ice temperature calibration was performed for each temperature sensor prior to the test. Twenty-four temperature data points were recorded for estimate at one-minute intervals and stored in IoT platform database and personal computer database. Indoor solar radiation sensor adopted HSTL-FSDJY type with a measurement range of $0\sim1500$ W·m$^{-2}$. Solar radiation data were acquired at one-minute intervals and stored in the IoT platform database. Outdoor meteorological data were collected and stored in outdoor weather stations, and it is a small automatic weather station HOBO U30 produced by Onset in the United States. Data recording interval was 10 min. The total solar radiation sensor had a measurement range of $0\sim1280$ W·m$^{-2}$ and an accuracy of $\pm10$ W·m$^{-2}$. The temperature sensor had a range of $-40\sim100$ °C and an accuracy of $\pm0.2$ °C. Otherwise, observations were also stored at slightly different intervals. The M2 and M3 data acquisition interval was 1 min, whereas the M1 and M4 data acquisition interval for supplementary analysis was 10 min. The data were stored in the IoT platform database (Figure 4b).

The test period was from December 2021 to February 2022. The crop cultivated in the greenhouse during the test was chilli. Due to the cultivation matrix and management problems, the growth of chilli was stagnant. Therefore, it can be assumed that there is no crop influence in the ideal state. The ventilation mode used in the greenhouse was natural ventilation, with lower vents and rear slope vents. In sunny days, the insulation cover was

usually opened between 8:00 and 9:00 every morning and closed between 16:00 and 17:00 every evening. In cloudy and snowy days, the time of opening the insulation cover was postponed according to the outdoor and indoor temperature gradient. The specific opening and closing times were based on the experience of the planting operator and the specific temperature in the greenhouse.

## 3. Results

### 3.1. Efficiency Analysis of Different Data Transmission Modes

The time interval between TDRMS and IoT platform data acquisition was 1 min. In theory, 1440 data were collected per data point every day. After removing errors due to program closing caused by human factors (active shut down program) and data missing caused by power outage in the greenhouse, the efficiency of CN and virtual LAN data acquisition technology used in IoT platform was analyzed. The data were preprocessed before calculating data efficiency and redundant data in the two methods were deleted to avoid distorting data qualified rate.

Data collected for nine consecutive days from 31 December 2021 to 8 January 2022 were selected for analysis. These data are shown in Table 3. The average data efficiency of CN technology transmission on IoT platform was 96.60%, whereas that of VLAN technology transmission was 99.24%.

**Table 3.** Analysis of valid data transmitted by two transmission methods.

| Date | Data Transmission Method | | | | | |
| | Cellular Network Technology Transmission | | | Virtual LAN Technology Transmission | | |
| | Valid Amount of Data/Piece * | Total Data /Piece | Data Validity Rate/% | Valid Amount of Data/Piece | Total Data/Piece | Data Validity Rate/% |
|---|---|---|---|---|---|---|
| 2021-12-31 | 34,152 | 34,560 | 98.82 | 34,332 | 34,560 | 99.34 |
| 2022-1-1 | 34,104 | 34,560 | 98.68 | 34,308 | 34,560 | 99.27 |
| 2022-1-2 | 33,744 | 34,560 | 97.64 | 34,428 | 34,560 | 99.62 |
| 2022-1-3 | 32,160 | 34,560 | 93.06 | 34,452 | 34,560 | 99.69 |
| 2022-1-4 | 33,360 | 34,560 | 96.53 | 34,452 | 34,560 | 99.69 |
| 2022-1-5 | 32,208 | 34,560 | 93.19 | 34,266 | 34,560 | 99.15 |
| 2022-1-6 | 33,648 | 34,560 | 97.36 | 33,996 | 34,560 | 98.37 |
| 2022-1-7 | 33,120 | 34,560 | 95.83 | 34,224 | 34,560 | 99.03 |
| 2022-1-8 | 33,960 | 34,560 | 98.26 | 34,212 | 34,560 | 98.99 |

\* 24 data points were transmitted each minute and collected 1440 times a day. Valid data refer to values that can be used for data analysis after the abnormal values (repeated data and 0) were excluded from normal data received.

Analysis results showed that both CN technology transmission and VLAN technology transmission on IoT platform met data efficiency demands. However, data efficiency of VLAN transmission mode was 2.64% higher than that of CN technology transmission on IoT platform. Variance analysis of effective data showed that CN technology transmission and VLAN technology transmission had data efficiency variance of 4.80% and 0.18%, respectively. Further, VLAN technology transmission showed better data stability than CN technology transmission on IoT platform.

### 3.2. Temperature Estimate Effect Analysis

The temperature interpolation estimate used data from seven consecutive days from 28 December 2021 to 3 January 2022, including typically sunny days and cloudy days. The 15.0 m × 10.0 m area was divided into 60 × 60 grids using the interpolation estimate module. Grid coordinates of the estimate points were determined according to physical coordinates of the observation points. Linear fitting analysis was performed to determine the difference between observed values and estimated values. The observed values were the temperature data of M2 and M3. Before interpolation estimate, data were preprocessed,

and all zero values were removed. To identify and remove abnormal data, the array was tested with the following triple standard deviation method (the Laida criterion):

$$x_{\text{normal value}} \in (\overline{x} - 3\sigma, \overline{x} + 3\sigma) \tag{6}$$

In the formula, x and σ are the average value standard deviation of this dataset.

After the data were preprocessed, linear fitting analysis was performed using data points from 17,647 groups. The fitting results are shown in Figure 5. Results show that the average relative error between estimated value and observed value, average absolute error, root mean square error, and determination coefficient were 0.12 °C, 0.42 °C, 0.56 °C, and 0.9964, respectively.

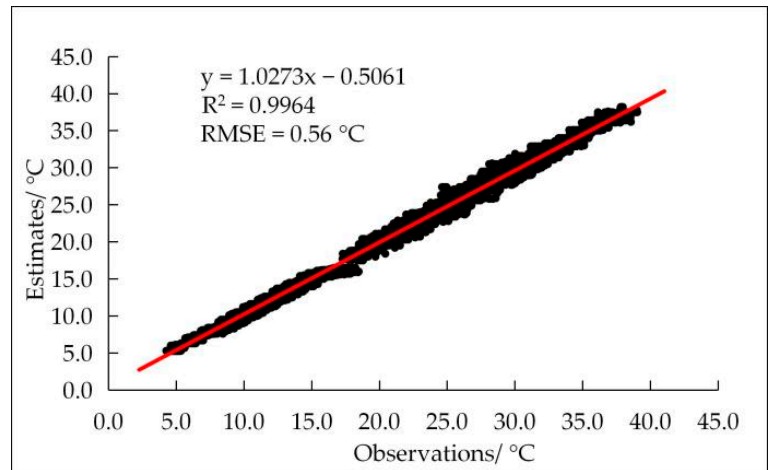

**Figure 5.** Linear fitting analysis chart of estimated and observed values. The observed data are the temperature data of M2 and M3. The red curve is the linear fitting curve, and the black dots are the observed and estimated values.

When original observed data values were analyzed during the test, it was found that temperature accuracy of data changed significantly on sunny days than on rainy days (Figure 6a). This indicated that the high accuracy and sensitivity of the Pt100 sensor could not be fully reflected in relatively small temperature fluctuations. Combined with the observation data acquisition module of other analog acquisition data condition analysis (the changes of solar radiation and relative humidity of the DAM in rainy days), under the condition of rainy weather, data change was not obvious due to the observation data acquisition module own condition limit (long service life, module hardware aging, etc.), resulting in the situation. Therefore, based on original observation points, we solved this problem by replacing DAM and supplementing linear fitting analysis of estimated data and measured data with M1 and M4 observation points (Figure 6b). The reason for the large temperature difference between M4 and M1 was that the two temperature measuring points were located on the east and west sides of the greenhouse. M2 and M3 were located near the middle of the greenhouse, so the temperature difference was small. The distribution of temperature measurement points is shown in Figure 2b.

The supplementary data were acquired from 8 February 2022 to 15 February 2022 at an interval of 10 min. The acquisition interval was increased to reduce iteration steps for large amounts of data and improve operational efficiency of the program. Linear fitting analysis was performed on the 1987 group of data points collected within eight consecutive days. Fitting results are shown in Figure 7. For this analysis, the average relative error between the estimated value and the observed value, average absolute error, root mean square, and error determination coefficient were −0.24 °C, 0.34 °C, 0.46 °C, and 0.9972, respectively.

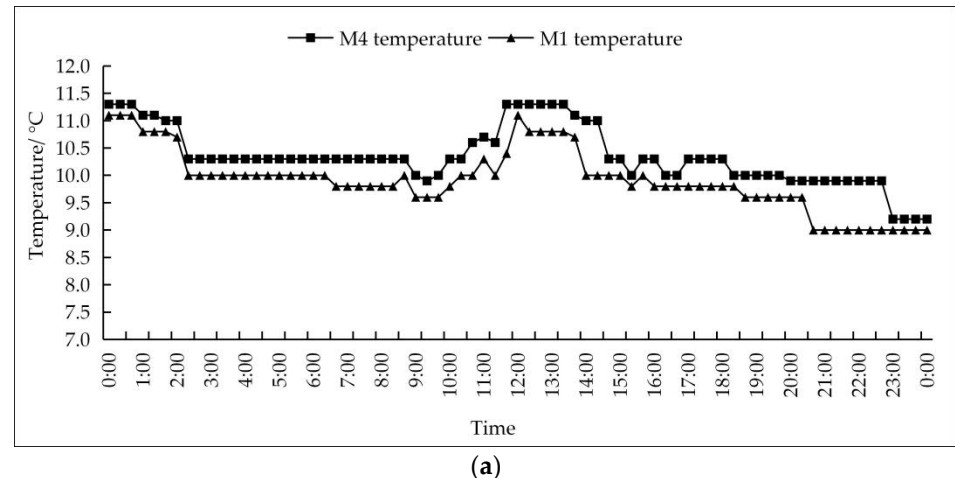

(**a**)

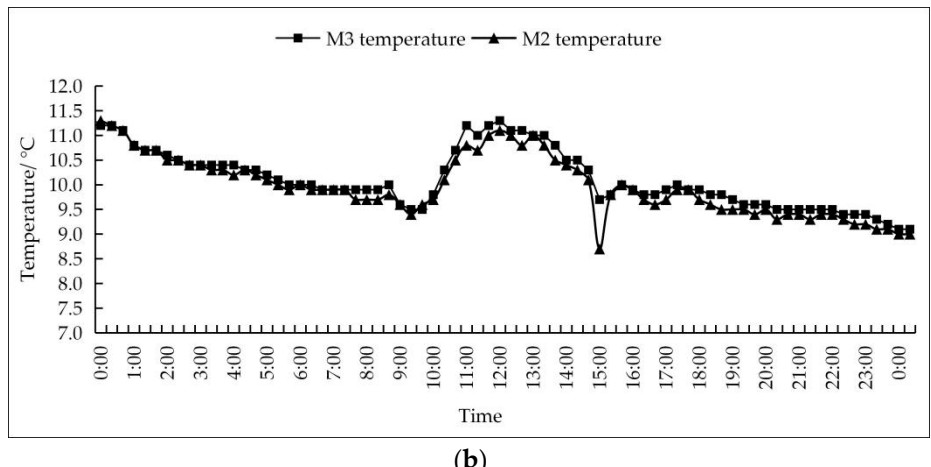

(**b**)

**Figure 6.** Temperature changes in greenhouses during rainy days on 16 February 2022. (**a**) Temperature changes of M2 and M3 measuring points; (**b**) Temperature changes of M1 and M4 measuring points.

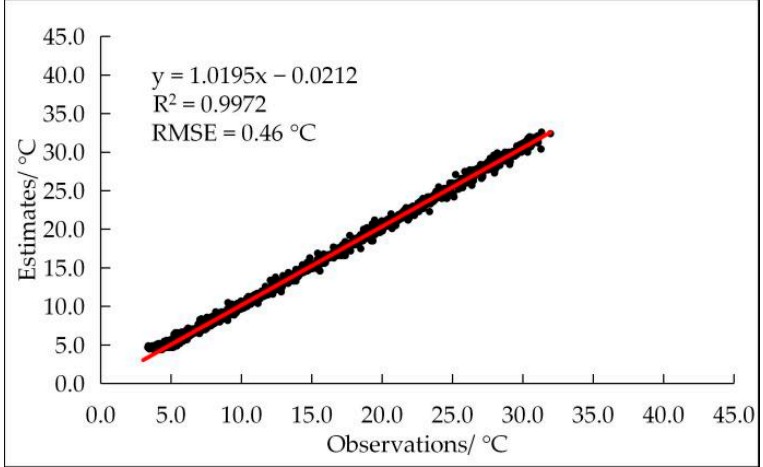

**Figure 7.** Linear fitting analysis chart of estimated and observed values. The observed data are the temperature data of M1 and M4.

For analysis of data from two consecutive acquisition cycles, the average relative error between the estimated value and the observed value, average absolute error, root mean square error, and $R^2$ were $-0.18$ °C, 0.38 °C, 0.51 °C, and >0.99, respectively. This method

yielded ideal fitting results for temperature distribution in the solar greenhouse, including large determination coefficient and small root mean square error. To some extent, the results accurately reflect actual distribution of temperature in the greenhouse, hence they are useful for real-time monitoring of greenhouse temperature.

### 3.3. Analysis of the Influence of Reducing Number of Sensors on the Accuracy of Temperature Estimate

From the analysis in Section 3.2, it was determined that Ordinary Kriging interpolation was feasible in temperature distribution estimate in solar greenhouse. Therefore, using original 24 temperature measuring points, an optimal number of sensors was obtained by reducing all sensors in the whole row or column regularly to reduce the cost of the system.

#### 3.3.1. Reduction of Four Temperature Sensors

There were four ways to reduce four temperature measuring points, namely, removing columns B, C, D, and E. After deleting the four data points, M2 and M3 data were analyzed by linear fitting with estimated values. The results are shown in Figure A2 (The fitting diagram is shown in Appendix A). The average relative error was $-0.19$ °C, the average absolute error was 0.44 °C, and the average root mean square error was 0.58 °C. The analysis data of the four ways are shown in Table 4.

**Table 4.** Linear correlation analysis results after reducing 4 sensors.

| Number of Sensors/ Piece | Reduce Sensor Position * | Mean Relative Error/°C | Mean Absolute Error/°C | Linear Fitting Curve | $R^2$ | RMSE/°C |
|---|---|---|---|---|---|---|
| 20 | B | −0.30 | 0.48 | y = 0.9872x + 0.4734 | 0.9955 | 0.62 |
| 20 | C | −0.09 | 0.41 | y = 0.9735x + 0.4665 | 0.9961 | 0.56 |
| 20 | D | −0.20 | 0.43 | y = 0.9800x + 0.4773 | 0.9962 | 0.56 |
| 20 | E | −0.18 | 0.42 | y = 0.9786x + 0.4817 | 0.9961 | 0.56 |

* Where B to E are column mark.

#### 3.3.2. Reduction of Six Temperature Sensors

There were two ways to reduce six temperature measuring points, namely, removing rows 2 and 3. After deleting the six data points, M2 and M3 data from two observation points and estimate value were analyzed by linear fitting. The results are shown in Figure A3. The average relative error, average absolute error, and average root mean square error were $-0.03$ °C, 0.46 °C, and 0.59 °C. Results of the analysis using the two ways are shown in Table 5.

**Table 5.** Linear correlation analysis results after reducing 6 sensors.

| Number of Sensors/ Piece | Reduce Sensor Position * | Mean Relative Error/°C | Mean Absolute Error/°C | Linear Fitting Curve | $R^2$ | RMSE/°C |
|---|---|---|---|---|---|---|
| 18 | ② | 0.17 | 0.40 | y = 0.9815x + 0.4279 | 0.9962 | 0.54 |
| 18 | ③ | −0.22 | 0.51 | y = 0.9706x + 0.6345 | 0.9952 | 0.64 |

* Where ② to ③ are row marks.

#### 3.3.3. Reduction of Eight Temperature Sensors

There were six ways to reduce the eight temperature measuring points, namely, removing BC, BD, BE, CD, CE, and DE. After removing eight data points, M2 and M3 data from two observation points and the estimate value were analyzed by linear fitting. The results are shown in Figure A4. The average relative error, average absolute error, and average root mean square error were $-0.24$ °C, 0.45 °C, and 0.57 °C, respectively. Results of the analysis using the six ways are shown in Table 6.

**Table 6.** Linear correlation analysis results after reducing 8 sensors.

| Number of Sensors/ Piece | Reduce Sensor Position * | Mean Relative Error/°C | Mean Absolute Error/°C | Linear Fitting Curve | R² | RMSE/°C |
|---|---|---|---|---|---|---|
| 16 | B,C | −0.30 | 0.5 | y = 0.9980x + 0.3276 | 0.9943 | 0.68 |
| 16 | B,D | −0.33 | 0.48 | y = 0.9985x + 0.3547 | 0.9956 | 0.63 |
| 16 | B,E | −0.11 | 0.38 | y = 0.9886x + 0.2807 | 0.9965 | 0.38 |
| 16 | C,D | −0.16 | 0.40 | y = 0.9854x + 0.3635 | 0.9961 | 0.54 |
| 16 | C,E | −0.18 | 0.40 | y = 0.9852x + 0.3897 | 0.9963 | 0.53 |
| 16 | D,E | −0.37 | 0.51 | y = 1.0016x + 0.3456 | 0.9949 | 0.68 |

* Where the letters represent column marks.

### 3.3.4. Reduction of Twelve Temperature Sensors

There were four ways to reduce 12 temperature measuring points, namely, removing columns BCD, BCE, and CDE and rows 2 and 3. After removing 12 data points, M2 and M3 data from two observation points and the estimate value were analyzed by linear fitting. The results are shown in Figure A5. The average relative error, average absolute error and average root mean square error were −0.38 °C, 0.54 °C, and 0.73 °C, respectively. Analysis results using the four ways are shown in Table 7.

**Table 7.** Linear correlation analysis results after reducing 12 sensors.

| Number of Sensors/ Piece | Reduce Sensor Position * | Mean Relative Error /°C | Mean Absolute Error /°C | Linear Fitting Curve | R² | RMSE/°C |
|---|---|---|---|---|---|---|
| 12 | ②,③ | −0.46 | 0.59 | y = 0.9964x + 0.5047 | 0.9952 | 0.72 |
| 12 | B,C,D | −0.33 | 0.52 | y = 1.0143x + 0.1338 | 0.9931 | 0.76 |
| 12 | B,C,E | −0.33 | 0.54 | y = 1.0015x + 0.3089 | 0.9932 | 0.74 |
| 12 | C,D,E | −0.38 | 0.51 | y = 1.0135x + 0.3089 | 0.9945 | 0.71 |

* In the second column, letters represent column marks while numbers represent row marks.

### 3.3.5. Reduction of Sixteen Temperature Sensors

There was one way to reduce 16 temperature measuring points, namely, removing columns B, C, D, and E. After deleting 16 data points, M2 and M3 data from two observation points and the estimate value were analyzed by linear fitting. The results are shown in Figure A6. The relative error, absolute error, and root mean square error were −0.57 °C, 0.67 °C, and 0.92 °C. Analysis results using the four ways are shown in Table 8.

**Table 8.** Linear correlation analysis results after reducing 16 sensors.

| Number of Sensors/ Piece | Reduce Sensor Position * | Mean Relative Error /°C | Mean Absolute Error /°C | Linear Fitting Curve | R² | RMSE/°C |
|---|---|---|---|---|---|---|
| 8 | B,C,D,E | −0.57 | 0.67 | y = 1.0413x + 0.0096 | 0.9936 | 0.92 |

* Where the letters represent column marks.

After reducing the number of sensors, the relationship between the relative error, absolute error, root mean square error, and the number of sensors was analyzed. The results are shown in Figure 8. The number of sensors decreased from 24 to 8, whereas the absolute value of the average relative error gradually increased, suggesting that the overall estimated value was smaller than the observed value. The average absolute error and root mean square error gradually increased, but the difference in number of sensors between 16 and 24 was not significant. As the number of sensors continued to decrease, the average relative error, the average absolute error, and the average root mean square error changed significantly. Therefore, we speculated that when the number of sensors decreases to 12~16, the average absolute error, root mean square error, and determination coefficient would be 0.40~0.60 °C, 0.60~0.80 °C, and R² > 0.99, respectively. These values meet the good estimate results of small root mean square error and large determination

coefficient. Therefore, 12–16 sensors could meet the needs of estimate and provide uniform sensor distribution that potentially yields better estimate effect.

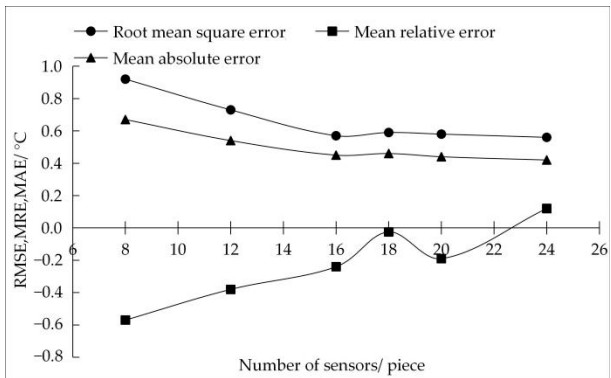

**Figure 8.** ▲ is the average of the mean absolute error after reducing the same number of sensors in different ways, ■ is the average of the mean relative error, and ● is the average of the root mean square error.

*3.4. Spatiotemporal Variation of Temperature Distribution in Modular Earth Wall Solar Greenhouse*

Monitoring temperature in solar greenhouse using TDRMS showed that the distribution of daytime temperature in greenhouse exhibited a strong spatial and temporal distribution in sunny winter. Therefore, the relationship between indoor and outdoor solar radiation, temperature, time, and space on typical sunny days was analyzed. The temperature data on the east and west sides of the greenhouse (Figure 9a) were collected at M2 and M3 temperature measuring points. The outdoor temperature and solar radiation data were from meteorological stations. The indoor solar radiation data were collected using indoor solar radiation sensors. In Figure 9b, the temperature data of the east and west sides were collected with two temperature measuring points: M1 and M4. In both cases, the parameters were the same. As shown in Figure 9a,b, the temperature on the east and west sides started to rise after the insulation was opened at around 9:00. Since the west side was first exposed to the sun, temporal and spatial temperature differences were observed on the east and west sides. The temperature on the west side reached the maximum value two hours earlier than that on the east side. In addition, the maximum temperature value was reached in the west side one hour earlier than that in the interior and exterior solar radiation. Finally, the maximum temperature value in the east was reached one hour later than that in the interior and exterior solar radiation.

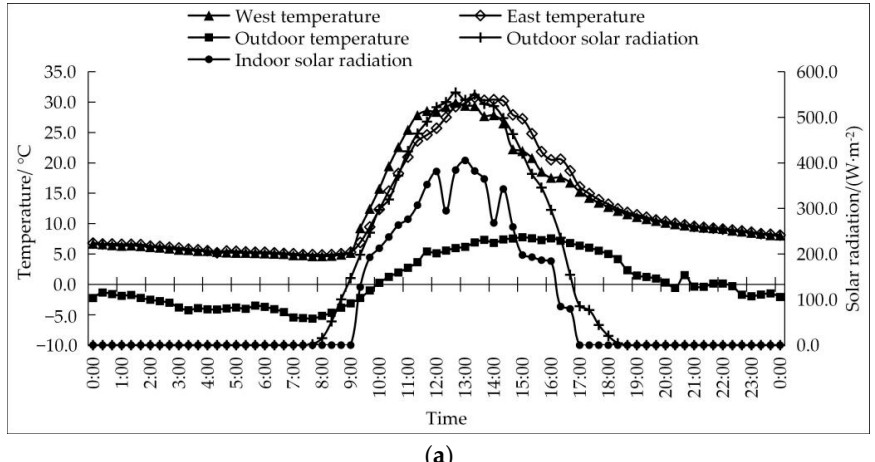

(a)

**Figure 9.** *Cont.*

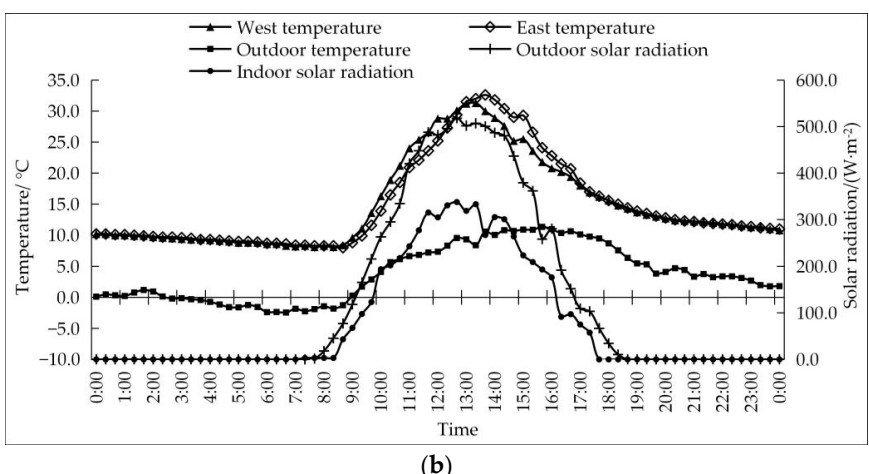

**Figure 9.** Typical sunny indoor and outdoor temperature, solar radiation temperature changes. (**a**) Data acquisition date is 9 February 2022; (**b**) Data acquisition date is 13 February 2022.

In Figure 10a,b, TDRMS was used to estimate temperature distributions at two time points in two typical winter sunny days of 9 February 2022 and 13 February 2022 in the greenhouse (More temperature distributions at time points are shown in Figure A7 of Appendix A). Temporal and spatial variations of temperature distributions begun to rapidly rise from south and west sides of the greenhouse. Since solar radiation was first received near the west wall, the temperature rapidly increased. With changes in solar orientation, the rapid warming area of the greenhouse gradually moved eastwards and northwards, and the increase in temperature on the east side was accelerated. During changes in greenhouse temperatures, temperature changes at the position close to the north wall in the greenhouse was always slow, which was attributed to heat preservation. In the winter, heat preservation was not fully opened, thus heat preservation was blocked by sunlight. Most of the north wall walls were unable to be irradiated, while heat preservation and heat storage functions of the wall were weakened, resulting in a slow temperature rise near the north wall.

Since the solar radiation sensor in the experimental greenhouse was located in the center of the greenhouse length direction, variations in indoor solar radiation were basically consistent with those of outdoor solar radiation. Analysis of the data showed that solar radiation distribution in the greenhouse obeys the same law as temperature distribution (Figure 9), according to temperature level distribution in the greenhouse and the relationship between outdoor temperature and solar radiation. The planting area close to the east and west walls was not exposed to sunlight for a long time. In winter, there was slow warming on eastern and rapid cooling on western, and the temperature distribution in the greenhouse was very uneven.

### 3.5. Calculation Speed Analysis of the Interpolation Estimate Program

To determine whether TDRMS can realize fast interpolation estimate of temperature distribution in the greenhouse, a time-consuming analysis of interpolation estimate calculation part was performed. Running time of the program was represented by reading the running time of multiple sets of data in the file. The time required for a single calculation was found to be about 50 ms (Table 9). Then, when various single-group data calculations were performed, the time was 50~150 ms, therefore, the time used to analyze the program in the data calculation part was 50~150 ms. With increasing single calculation data, the calculation time was bound to increase, but in estimate of greenhouse temperature demand, the operation requirements were met.

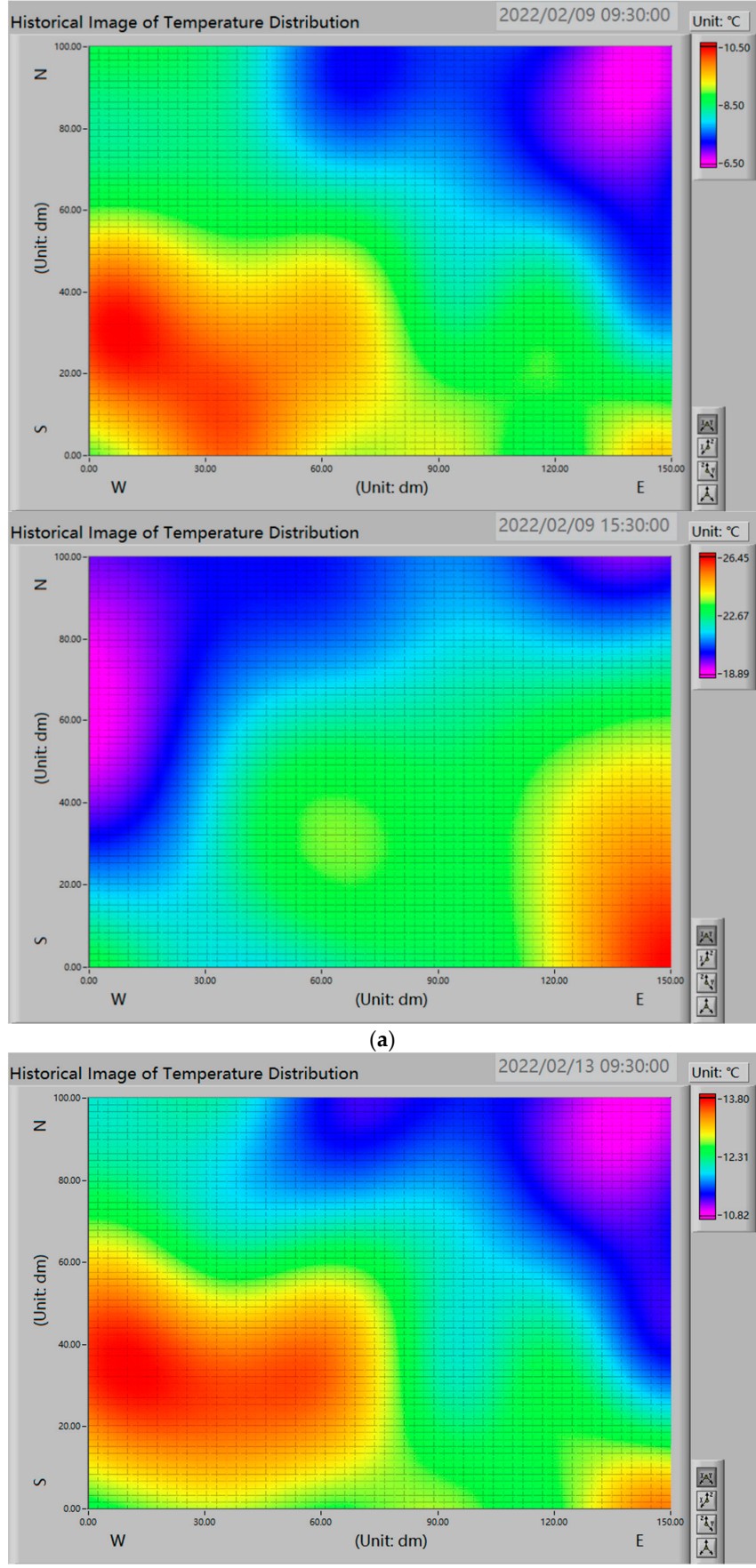

(**a**)

**Figure 10.** *Cont.*

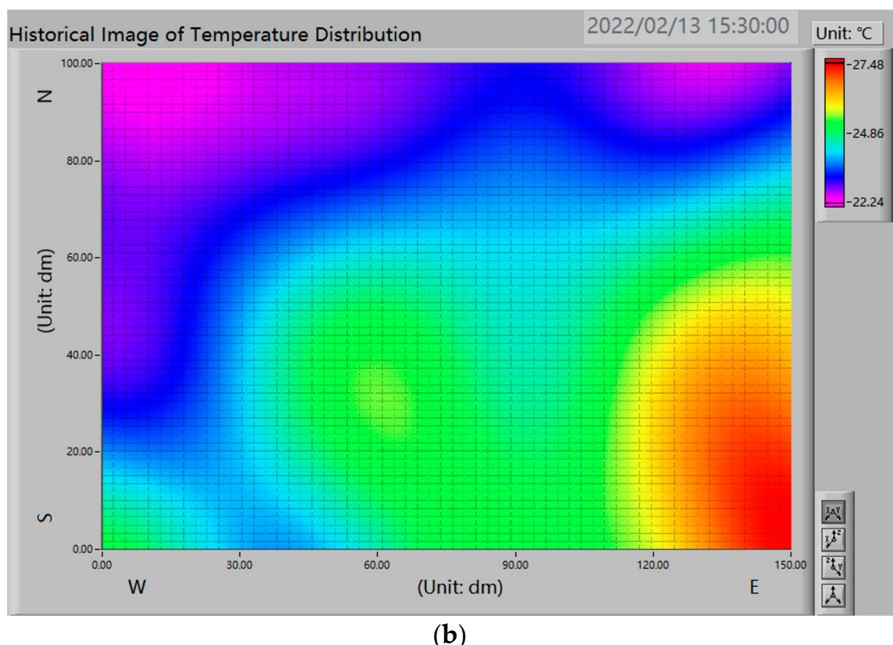

**(b)**

**Figure 10.** The horizontal temperature distribution of typical sunny days in winter was 9:30 and 15:30.
(**a**) Temperature distribution on 9 February 2022; (**b**) Temperature distribution on 13 February 2022.

**Table 9.** Time-consuming results of multi-group data estimate.

| Date | Total Data Number/Piece | Total Calculation Time/ms | Average Computation Time of Single Data/ms |
|---|---|---|---|
| 2021-12-28 | 1420 | 53,079 | 37.38 |
| 2021-12-29 | 1438 | 51,279 | 35.66 |
| 2021-12-30 | 1374 | 53,103 | 38.65 |
| 2021-12-31 | 1437 | 53,250 | 37.06 |
| 2022-01-01 | 1438 | 50,429 | 35.07 |
| 2022-01-02 | 1439 | 53,816 | 37.40 |
| 2022-01-03 | 1440 | 51,781 | 35.96 |
| 2022-02-08 | 1439 | 55,404 | 38.50 |
| 2022-02-09 | 1422 | 53,505 | 37.63 |
| 2022-02-10 * | 144 | 7818 | 54.29 |
| 2022-02-11 * | 144 | 7443 | 51.69 |
| 2022-02-12 | 1440 | 57,597 | 40.00 |
| 2022-02-13 | 1440 | 57,894 | 40.20 |
| 2022-02-14 | 1438 | 55,324 | 38.47 |
| 2022-02-15 | 1433 | 55,366 | 38.64 |

\* TDRMS do not run within two days of labeling, and the data are collected using the temperature data collected by IoT. Collection time interval was 10 min, while the total amount of data was 144 per day.

## 4. Discussion

### 4.1. Application Effects Evaluation of Real-Time Monitoring Systems for Temperature Distributions in Solar Greenhouse

Using the TDRMS data acquisition, it was found that interpolation accuracy and interpolation speed have good operation results. In the transmission mode, the use of virtual LAN technology to transmit data was less than the CN technology transmission protocol conversion process, thus, the VLAN technology transmission speed is faster and more stable; moreover, it can avoid errors in protocol conversion process, resulting in inaccurate data, therefore, VLAN transmission is better in data efficiency. With regards to the accuracy of interpolation, by analyzing the estimate results of 24 points of TDRMS, it is

established that estimate results had larger determination coefficients and smaller root mean square errors, which shows that estimate effects of TDRMS were good. In terms of interpolation speed, the running speed of TDRMS met the actual needs. Interpolation estimate results of TDRMS can quickly reflect temperature distributions of the corresponding positions in the greenhouse, especially during winter cultivation, which can quickly find the temperature anomaly area and reduce the risk of yield reduction to a certain extent. However, in error analysis, estimate results of TDRMS were generally smaller than the observed values, in contrast to findings from estimate results of Bojacá [21], which were generally higher than observed results. This analysis may be due to partial differences between algorithms in the R statistical analysis software and those in TDRMS. After data verification, observed and estimated values were collected for verification in this paper. In contrast to the cross-validation method adopted by Bojacá [21] and Zhang [22], the method used in this paper reduces the estimate error caused by deleting sensors at different positions.

### 4.2. Advantages and Disadvantages of TDRMS Compared with the IoT Platform

The TDRMS design is more personalized, and personalized services can be customized according to the needs of users; during data transmission, data is directly saved to the personal computer, avoiding data leakage, which can provide favorable support for movement or increase or decrease of relevant collection nodes, effectively reducing the amount of communication and improving the efficiency of network communication [28]; however, compared with the IoT platform, TDRMS has some limitations. The IoT platform has a wide range of users, more convenient operation, and a more beautiful interactive interface. However, the IoT platform is more inclined to data acquisition and storage platforms, has to comply with platform requirements, and has a strong limit on data storage period; TDRMS has more flexible access to data, has no format requirements, and has greater advantages in data processing and analysis.

After obtaining data, TDRMS can perform more complex secondary analyses, which can add or reduce system functionality according to requirements. For instance, TDRMS increases diversified services, such as suitability analysis of greenhouse microclimate and supplementary control of substances required for plant photosynthesis, such as carbon dioxide.

### 4.3. TDRMS Improvement Scheme and Research Prospects

During interpolation estimate, TDRMS relies on a large number of temperature data. In this experiment, the sensor optimization mainly considers the cost factor. The control variable is mainly the number of sensors, through the whole row and the whole column to reduce the number of sensors, to achieve a simple optimization effect. However, placing a reasonable number of sensors in important locations that can reflect the environmental characteristics can not only effectively reduce the cost of environmental monitoring, but also improve the data processing efficiency of the monitoring system. In addition, the effect of reducing the number of sensors on the accuracy of the results should be considered when optimizing the number of sensors. Studies have shown that the temperature differences between different locations, although they may only be 1 °C at the same time, can determine differential behaviours on plant growth and development [21]. The standard for controlling the number of sensors in this test is that the average absolute error between the estimated value and the real value is less than 0.80 °C. Compared with current clustering analysis to identify eigenvalues [29], based on interval possibility model and clustering regression distribution index to optimize sensor number and locations [30], there are various limitations in this experiment, which cannot reduce the number of sensors to the minimum number and find the most suitable measurement position. In the subsequent test process, it may increase the monitoring of eigenvalues in the greenhouse, and then optimize sensor

number and locations. Besides, TDRMS requires a high sensor accuracy, which entails both high-precision sensors and supporting data acquisition modules. In the early stages of this experiment, although data acquisition module types for M2 and M3 observations were suitable for Pt100 temperature sensors, due to limitations of the module, it is difficult to meet the demand for temperature data acquisition in cases of small temperature changes. Therefore, special instruments with a higher accuracy are needed. When TDRMS is used in large areas, it is necessary to increase the number of sensors and optimize the original algorithm to improve the operation speed. Therefore, for TDRMS, optimizing the number of sensors to select the appropriate sensor installation location to achieve better estimate results and reduce system operating costs are key factors for determining the application of the system to production.

In monitoring temperature distribution in the greenhouse, this system considers the distribution of the canopy position of the temperature. However, in the actual production process, the temperature above and below the plant canopy affects the temperature of the canopy. Therefore, it is necessary to monitor temperature distribution at different heights in real time.

The ordinary Kriging interpolation method can be used for interpolation estimate of environmental temperature and for interpolation estimate of environmental variables, such as precipitation. Distribution of humidity in the greenhouse is also uneven [31,32]. Therefore, this method can be extended to interpolation estimate of humidity in the greenhouse to analyze temporal and spatial distribution law of humidity in the greenhouse. Kriging interpolation plays an important role in spatial distribution analysis of soil nitrogen, phosphorus, and potassium [33]. This method can be used to estimate interpolation of large amounts of elements, such as nitrogen, phosphorus, and potassium, under soil cultivation mode in the greenhouse, which can provide some guidance and suggestions for rational planning of planting areas and rational fertilization.

In terms of ventilation, the existing research determines the ventilation mode and the number of fan openings [34], according to the estimate of greenhouse temperature field by the CFD software. Therefore, the visualization function of the real-time temperature monitoring system of solar greenhouses can provide some reference suggestions for formulation of a suitable ventilation strategy. Based on temperature distribution in the greenhouse and distribution positions of characteristic points, the appropriate number of ventilation fans and appropriate positions can be selected to determine the ventilation time and sizes of the natural vent. In the summer, the spray system plays an important role in cooling [35]. TDRMS was used to analyze correlations between humidity and temperature distributions in the greenhouse, and to determine whether it is necessary to cool the greenhouse through the spray system so as to avoid incorrect use of the spray system, leading to crop growth in a high humidity environment and hindering plant growth.

Through the interpolation principle of TDRMS, interpolation estimate of various environmental factors can be realized. According to interpolation estimate results of different environmental factors, the distribution of various environmental factors in the greenhouse can be analyzed, which can realize local regulation of indoor environmental factors, reduce energy consumption of solar greenhouse production, improve the output ratio, and reduce carbon emission of solar greenhouses.

## 5. Conclusions

TDRMS uses the virtual local area network technology to transmit temperature data in the greenhouse, which ensures data security to a certain extent, and then uses the geostatistical interpolation estimate method to realize real-time monitoring of temperature distribution in the greenhouse. Through preliminary verification of TDRMS in this experiment, it can be considered that TDRMS can provide a stable, fast, and accurate approach for real-time temperature monitoring of solar greenhouses. The following results were obtained from experimental data analysis:

(1) Data transmission efficiency of the VLAN technology in TDRMS is 2.64% higher than that of CN technology in the IoT platform, and the stability of VLAN transmission is better; during data transmission, data directly enters the private database, reducing the data leakage risk.

(2) In interpolation estimate using 24 temperature measuring points, the average relative error between the estimated values and the observed values is $-0.18\,^{\circ}\text{C}$, the average absolute error is $0.38\,^{\circ}\text{C}$, the root mean square error is $0.51\,^{\circ}\text{C}$, and the determination coefficient $R^2 > 0.99$, implying good estimate results. After optimizing the number of sensors, when the number of sensors is reduced to 12~16, the average absolute error is $0.40$~$0.60\,^{\circ}\text{C}$, the root mean square error is $0.60$~$0.80\,^{\circ}\text{C}$, and the determination coefficient $R^2$ is more than 0.99, which can still meet the needs for interpolation estimate.

(3) Combined with temperature distribution image analysis of TDRMS, temperature distribution inside the solar greenhouse has strong temporal and spatial distributions, and it increases from west to east and from south to north.

(4) TDRMS has a great advantage in estimate speed, and TDRMS realizes collection, processing, and real-time monitoring of greenhouse temperature distribution. Unlike computational fluid dynamics and neural network algorithms, TDRMS uses geostatistical interpolation estimates. The temperature interpolation calculation part has a fast calculation speed, and the time can be controlled in 50~150 ms, which realizes rapid monitoring. TDRMS not only guarantees the fast estimate speed, but also guarantees the good estimate effect, which meets the actual demand in production.

Additional details regarding this research are available in Supplementary Materials at https://doi.org/10.6084/m9.figshare.20171399.

**Supplementary Materials:** The following are available online at https://doi.org/10.6084/m9.figshare.20171399. File S1: 24 Sensors, File S2: Sensor Optimization 1, File S3: Sensor Optimization 2, File S4: TDRMS tem-perature acquisition raw data, File S5: VLAN and CN Data Validity; Program speed, File S6: Cross validation and model choose.

**Author Contributions:** Conceptualization, Y.C.; Formal analysis, S.Y., S.L. and X.C.; Methodology, S.Y. and X.L.; Resources, Y.C.; Software, S.Y.; Validation, Y.C.; Writing—original draft, S.Y.; Writing—review & editing, X.L., S.L. and X.C. All authors have read and agreed to the published version of the manuscript.

**Funding:** This research was funded by [Shaanxi Province Technological Innovation Guidance Special Project] grant number [No. 2021QFY08-02], and [Scientific & Technological Innovative Research Team of Shaanxi Province] grant number [No. 2021TD-34], and [Key Research & Development Project of Shaanxi Province] grant number [No. 2022ZDLNY03-02].

**Institutional Review Board Statement:** Not applicable.

**Informed Consent Statement:** Not applicable.

**Data Availability Statement:** The data presented in this study are available in supplementary materials and the following are available online at https://doi.org/10.6084/m9.figshare.20171399.

**Conflicts of Interest:** The authors declare no conflict of interest.

## Appendix A

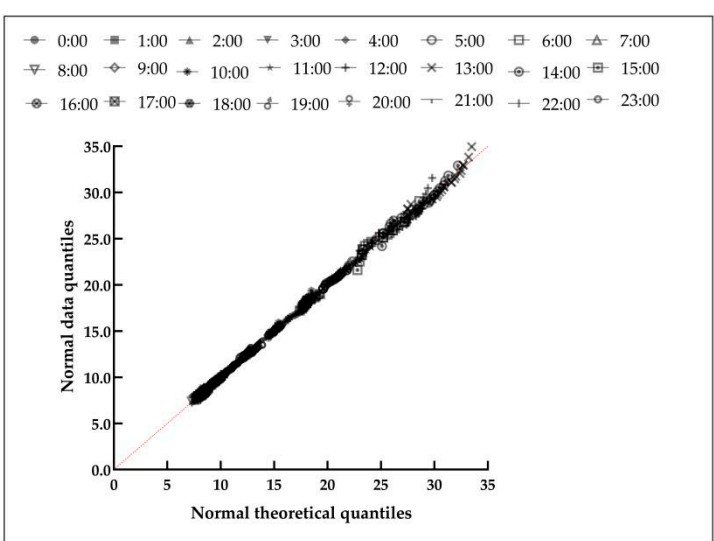

**Figure A1.** On 13 February 2022, normal distribution Q-Q diagram of temperature data at different observation points at different times.

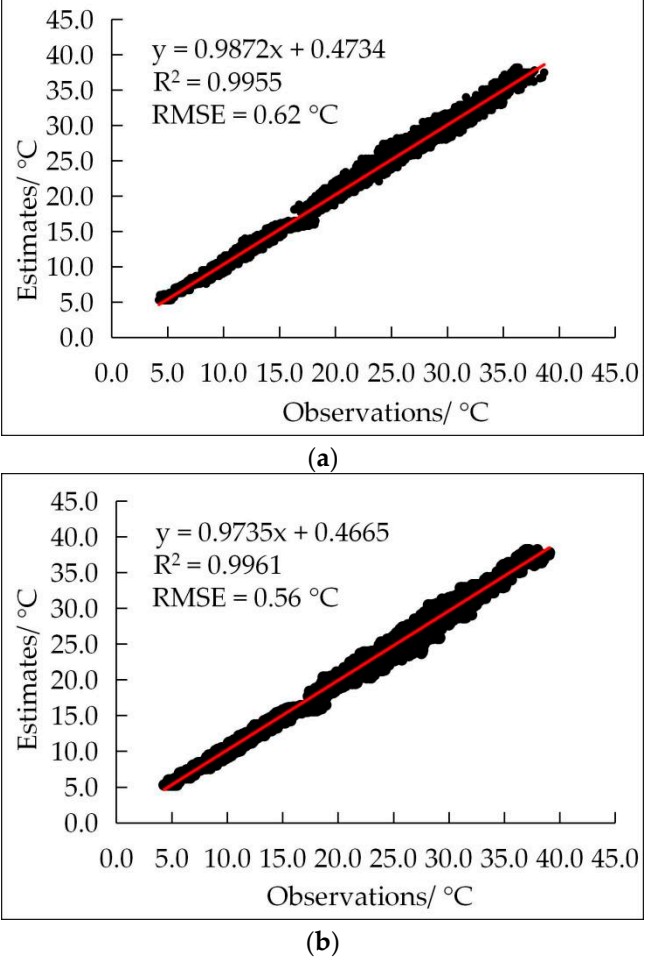

**Figure A2.** *Cont*.

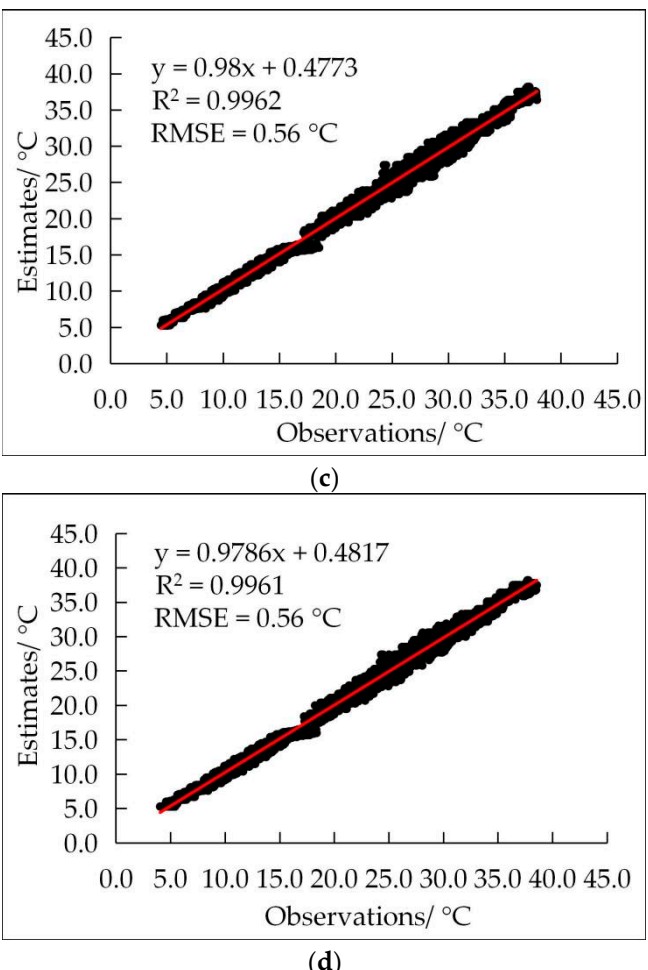

(**c**)

(**d**)

**Figure A2.** The linear fitting results of the observed and simulated values are reduced by 4 temperature sensors. (**a**) Reduced column B; (**b**) Reduced column C; (**c**) Reduced column D; (**d**) Reduced column E.

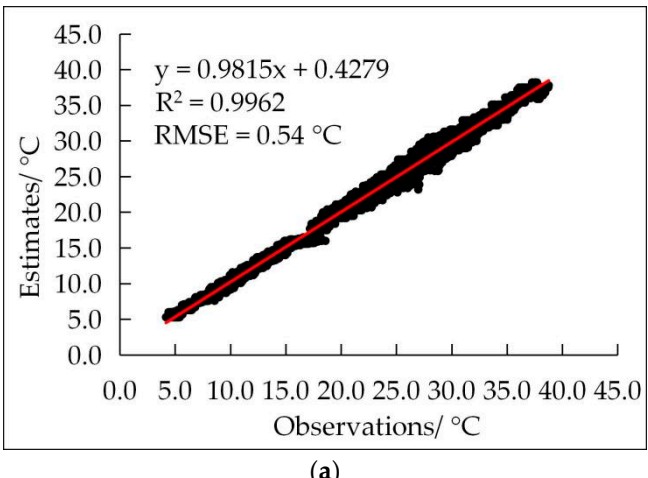

(**a**)

**Figure A3.** *Cont.*

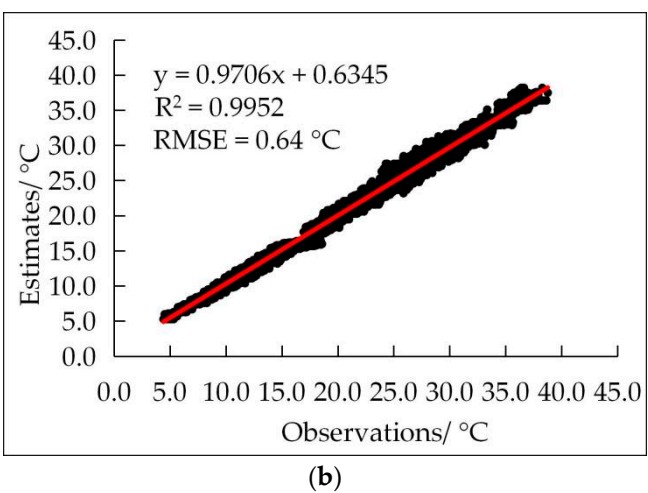

(**b**)

**Figure A3.** The linear fitting results of the observed and simulated values are reduced by 6 temperature sensors. (**a**) Reduced line ②; (**b**) Reduced line ③.

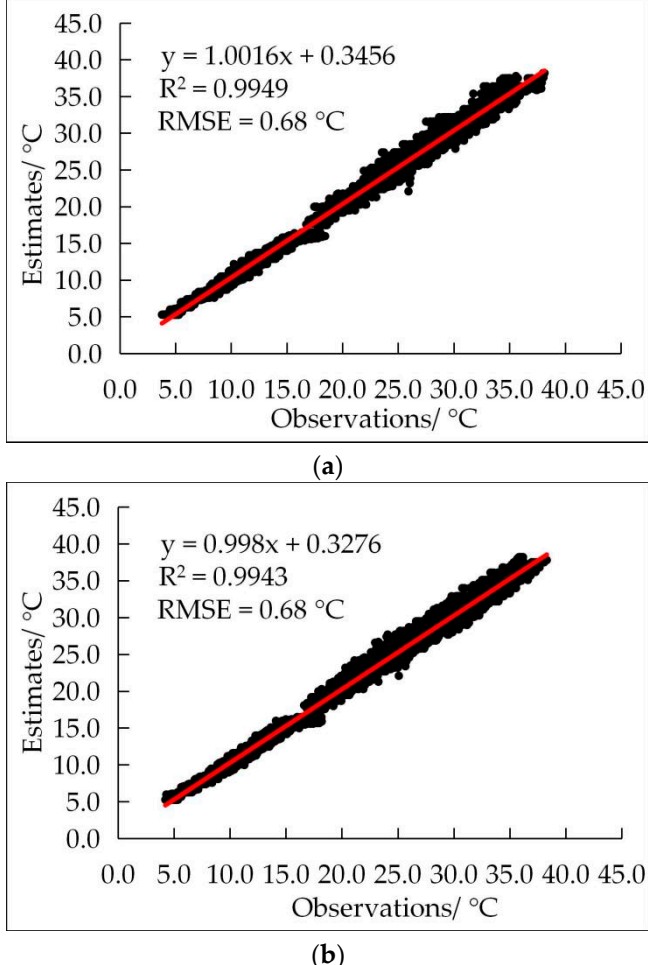

(**a**)

(**b**)

**Figure A4.** *Cont.*

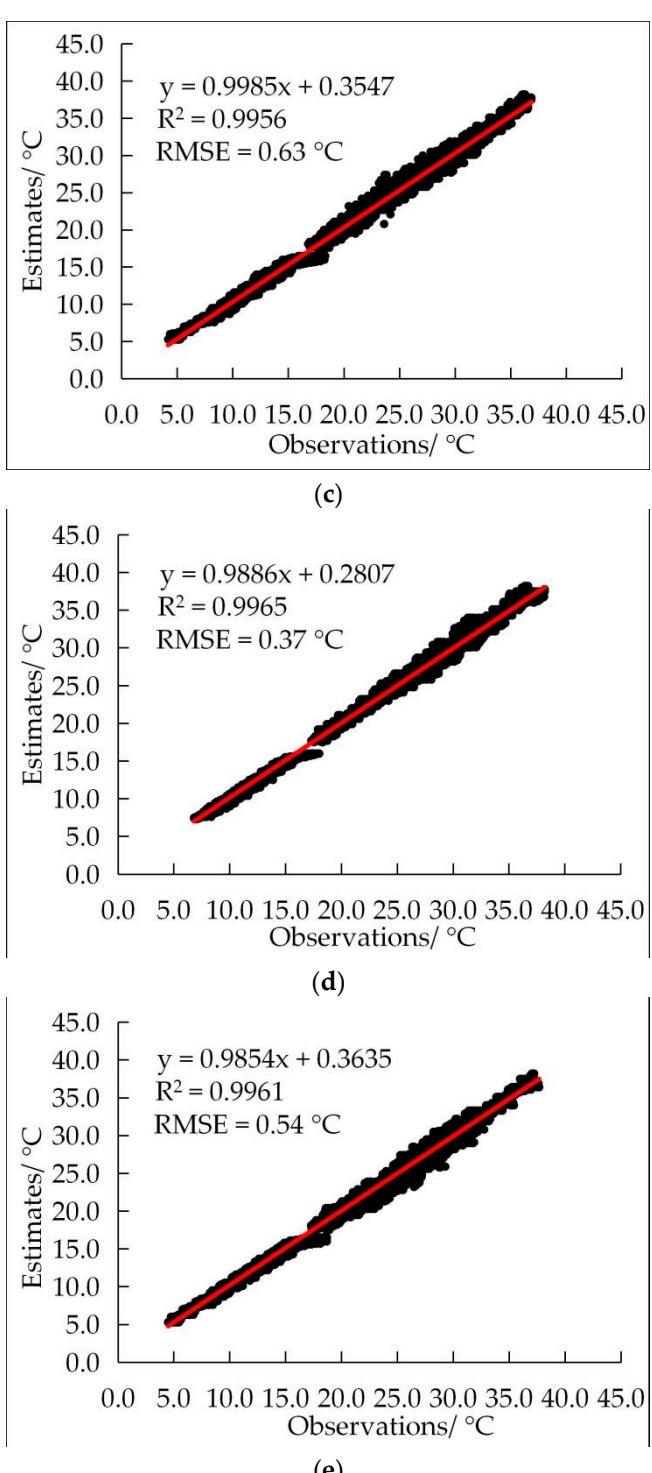

**Figure A4.** *Cont.*

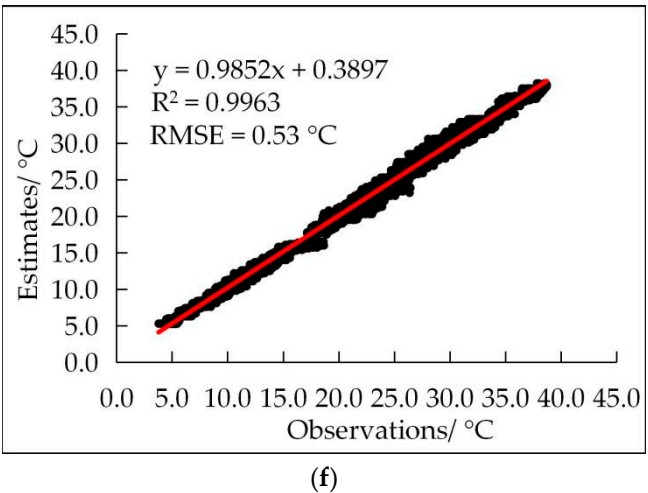

(**f**)

**Figure A4.** The linear fitting results of the observed and simulated values are reduced by 8 temperature sensors. (**a**) Reduced B and C columns; (**b**) Reduced B and D columns; (**c**) Reduced B and E columns; (**d**) Reduced C and D columns; (**e**) Reduced C and E columns; (**f**) Reduced D and E columns.

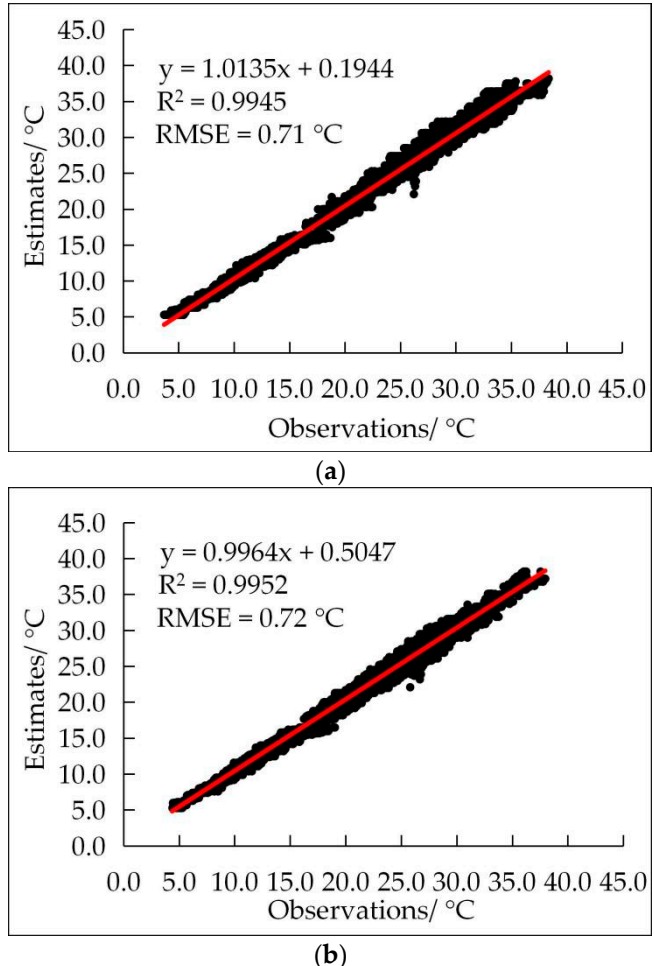

(**a**)

(**b**)

**Figure A5.** *Cont.*

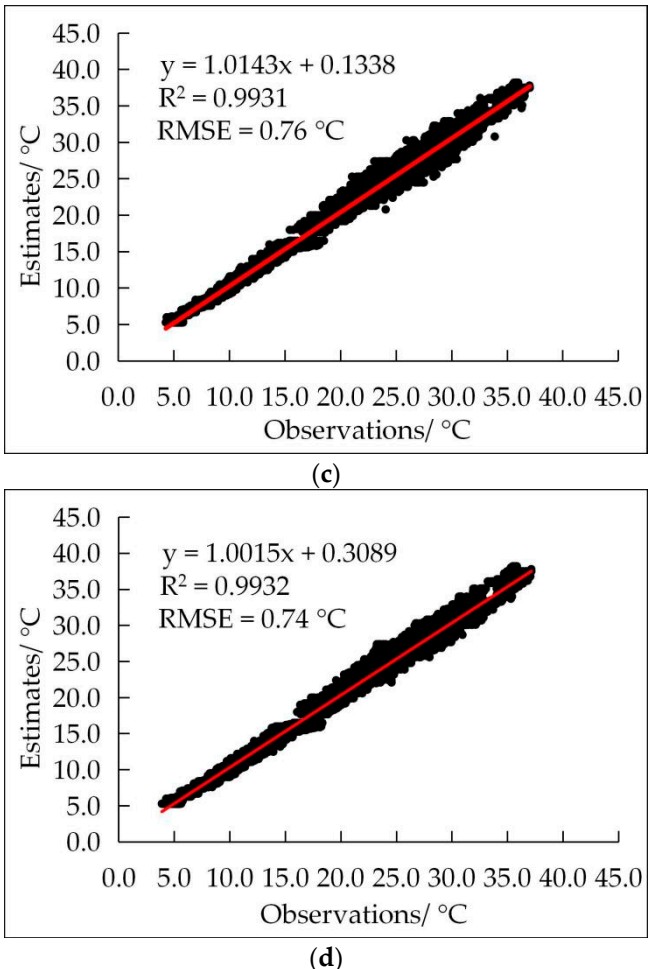

(**c**)

(**d**)

**Figure A5.** The linear fitting results of the observed and simulated values are reduced by 12 temperature sensors. (**a**) Reduced lines 2 and 3; (**b**) Reduced B, C, and D columns; (**c**) Reduced B, C, and E columns; (**d**) Reduced C, D, and E columns.

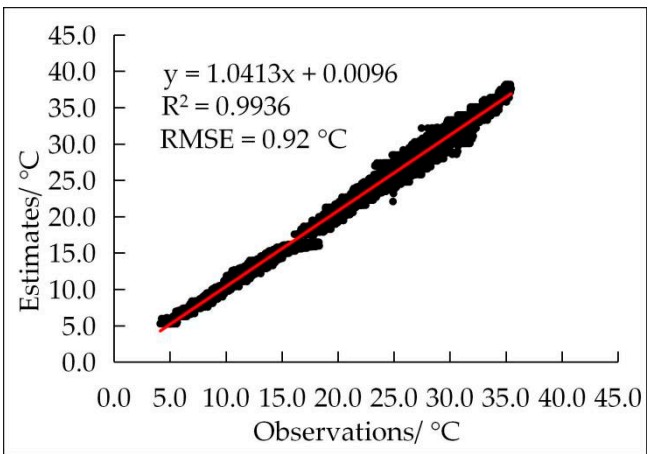

**Figure A6.** After reducing 16 temperature sensors, the linear fitting results between the observed and simulated values are reduced by B, C, D, and E columns.

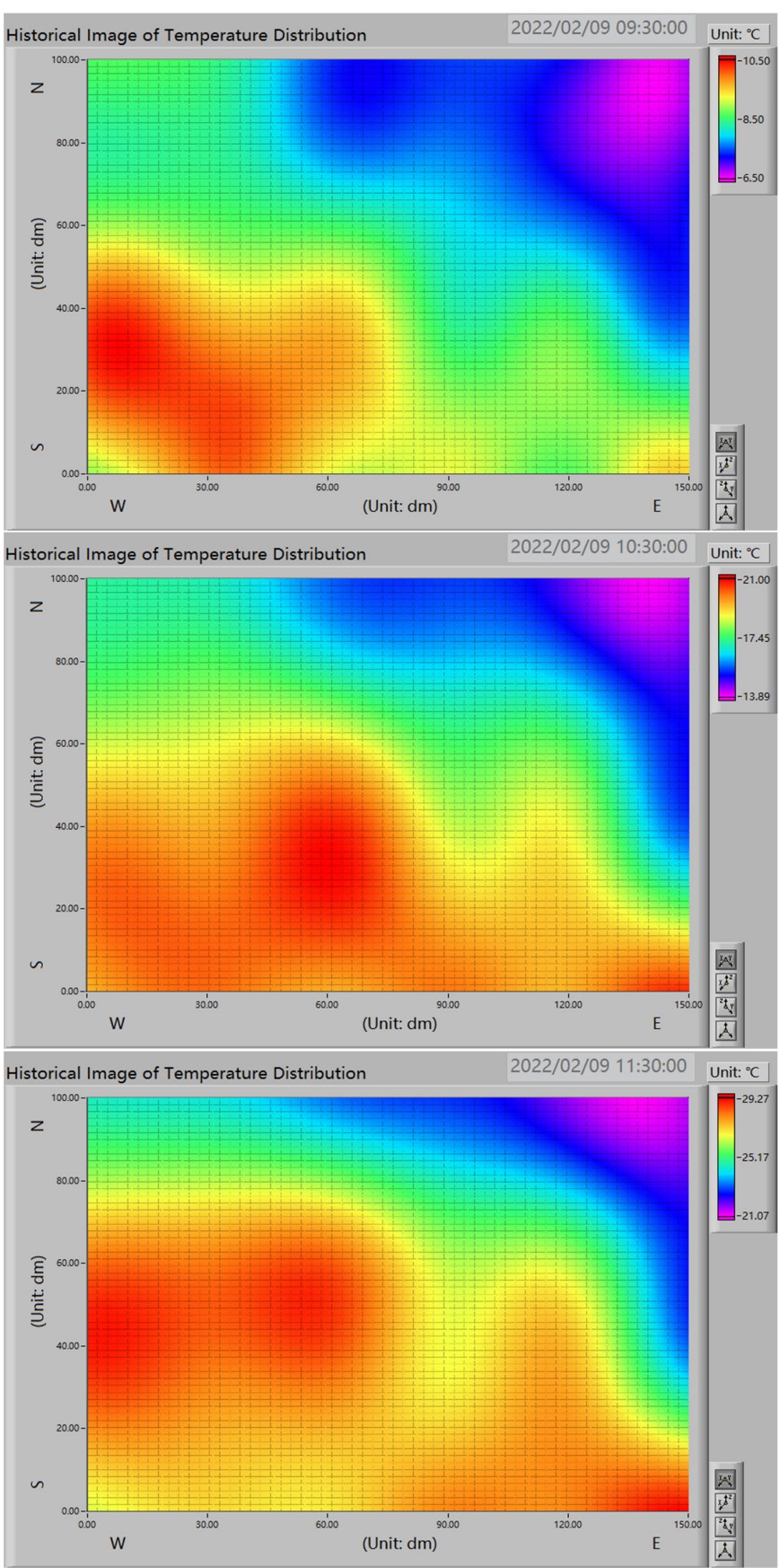

**Figure A7.** *Cont.*

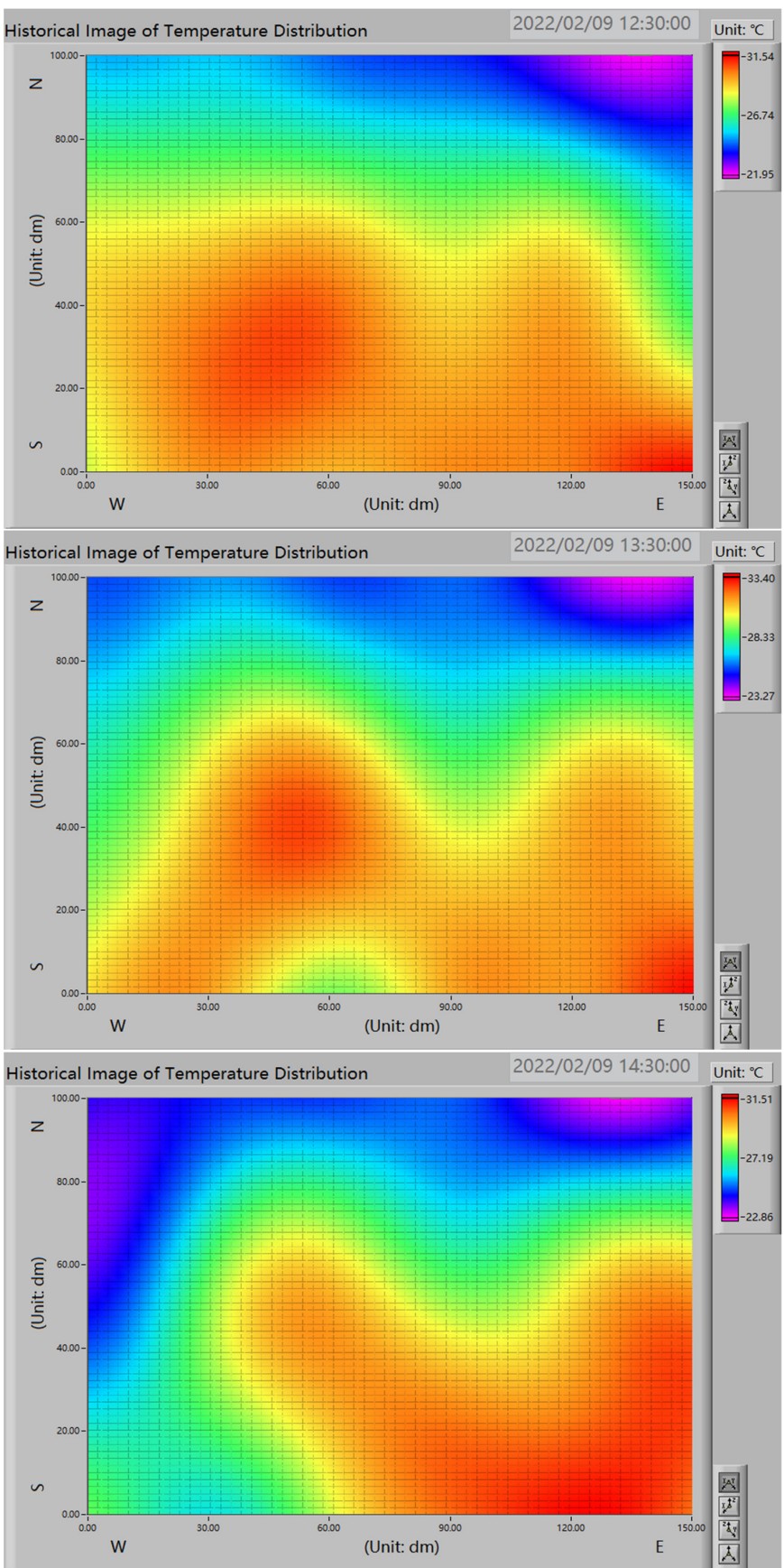

**Figure A7.** *Cont.*

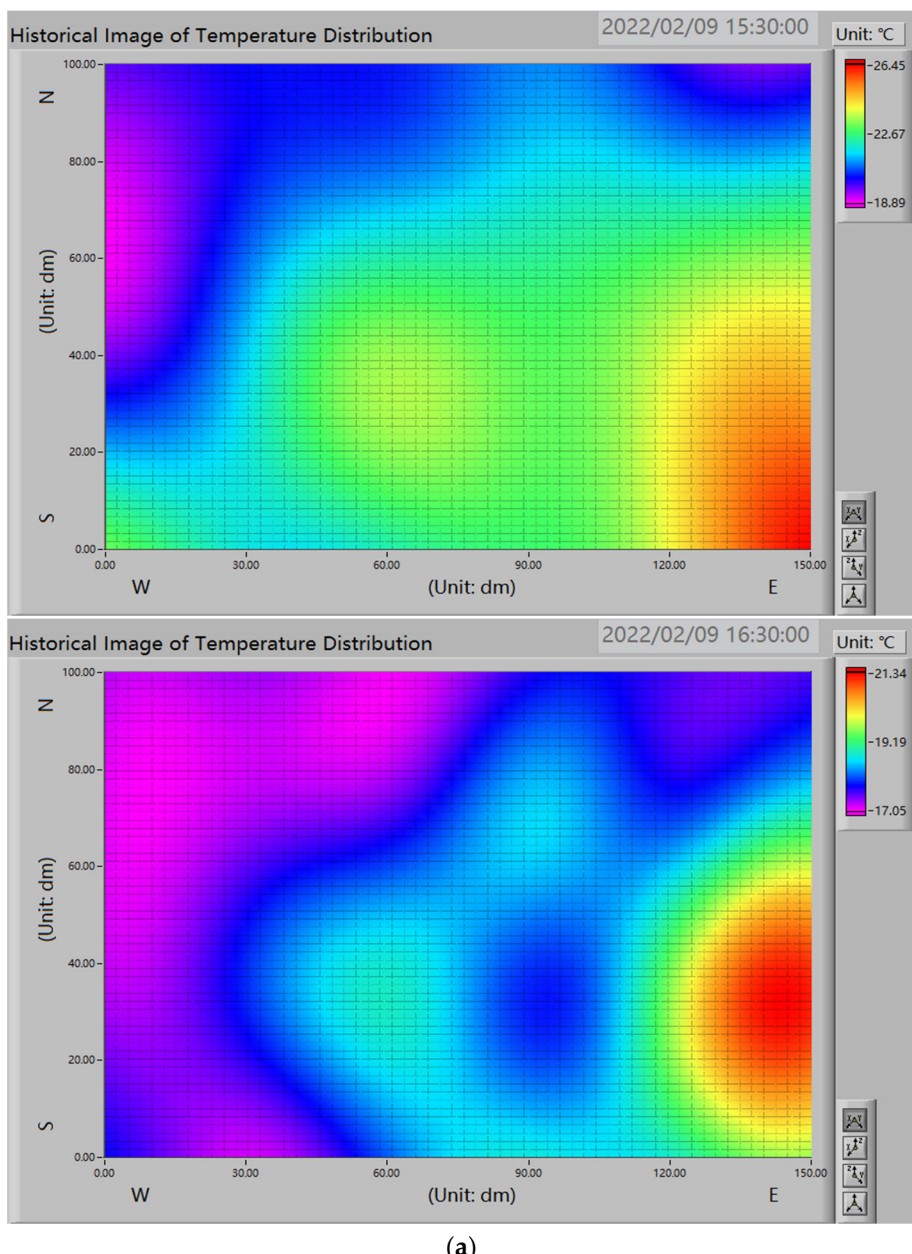

(**a**)

**Figure A7.** *Cont.*

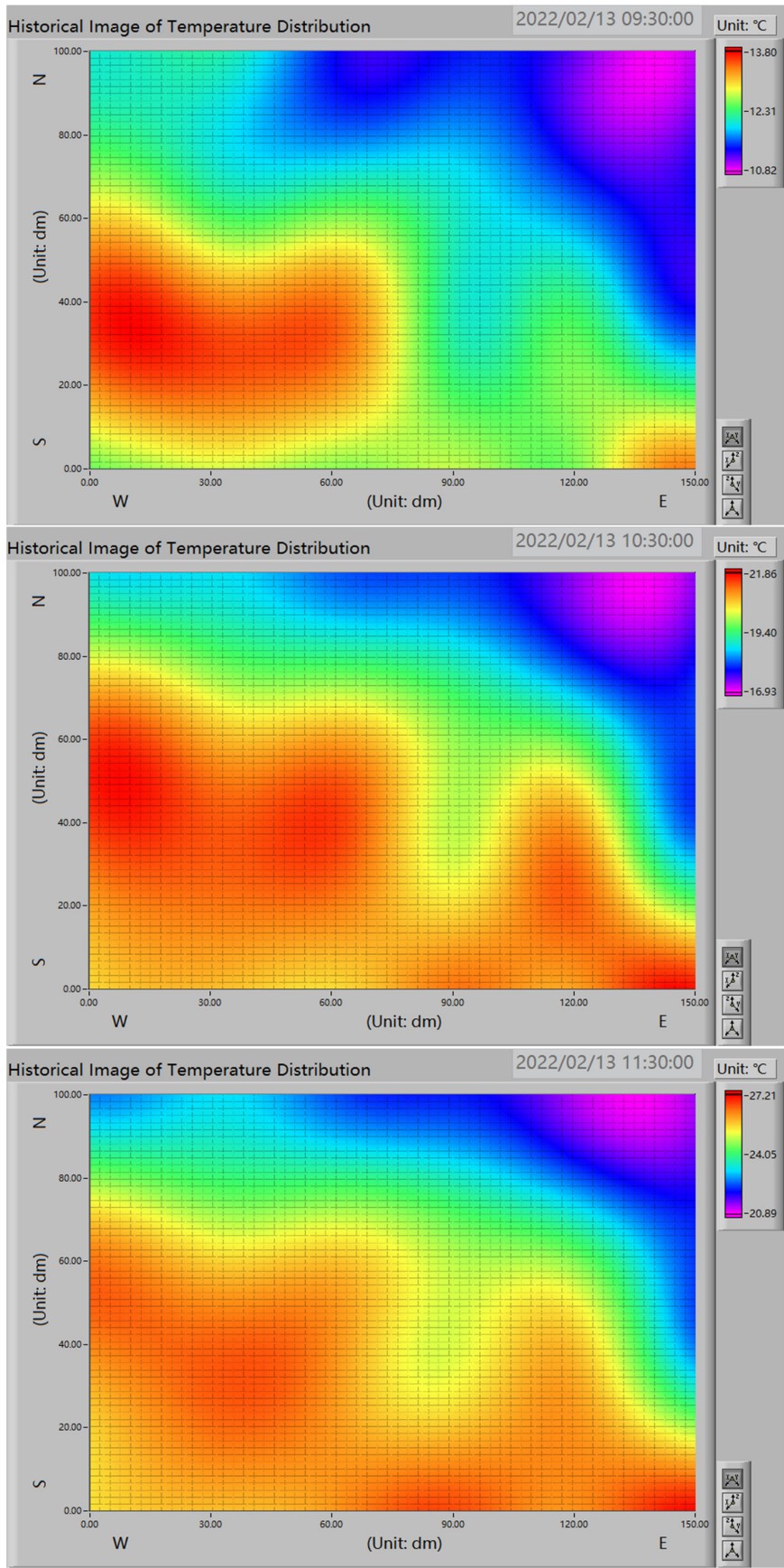

**Figure A7.** *Cont.*

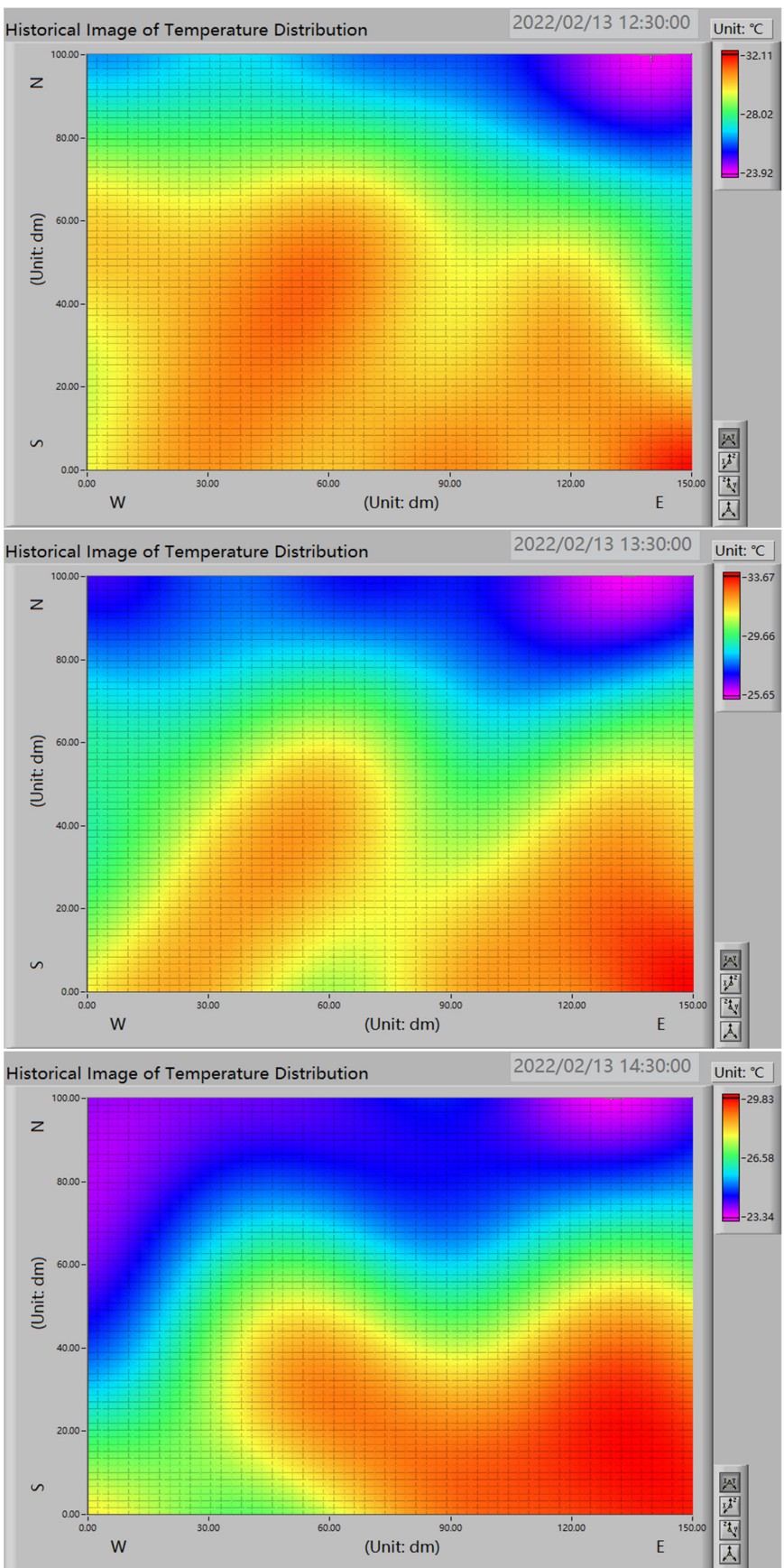

**Figure A7.** *Cont.*

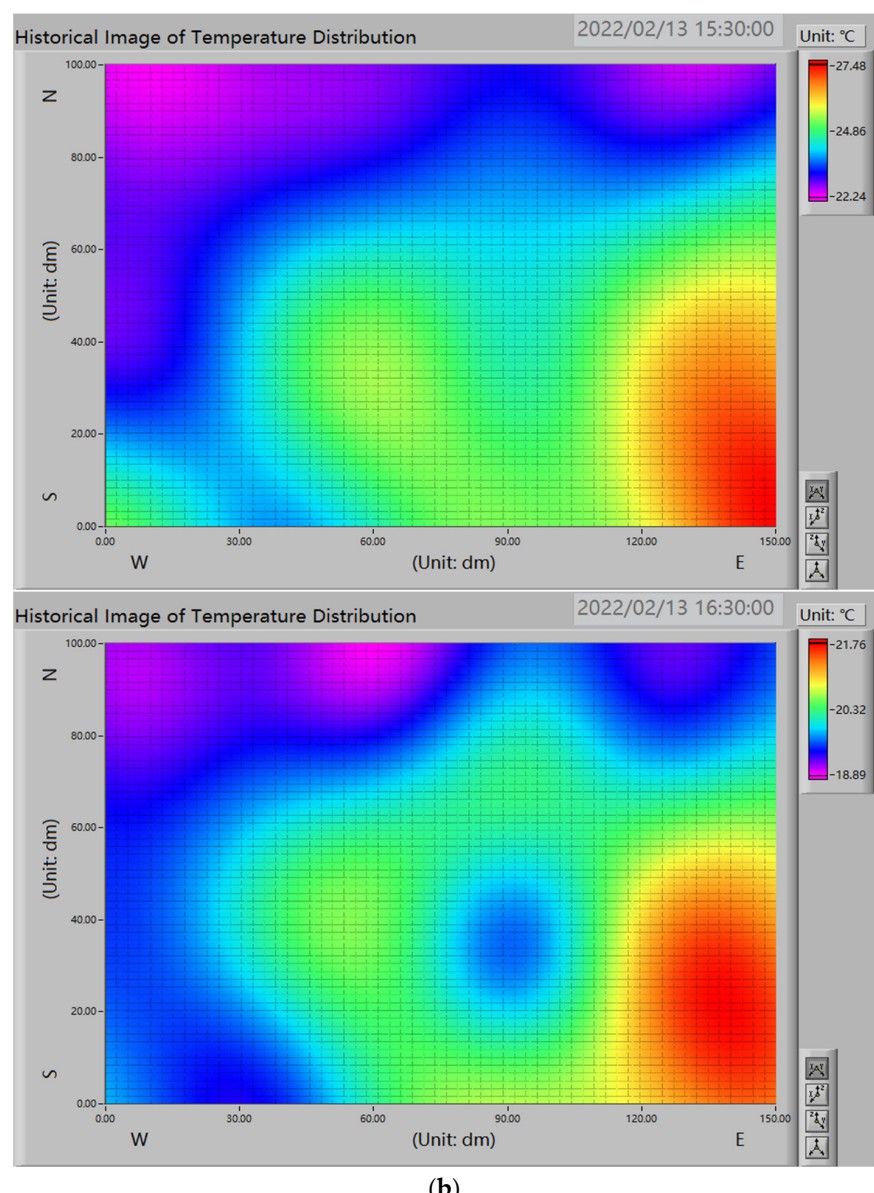

**(b)**

**Figure A7.** The horizontal temperature distribution of typical sunny days in winter was 9:30, 10:30, 11:30, 12:30, 13:30, 14:30, 15:30, and 16:30. (**a**) Temperature distribution on 9 February 2022; (**b**) Temperature distribution on 13 February 2022.

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
