# Peer review of "Real-Time Temperature Distribution Monitoring in Chinese Solar Greenhouse Using Virtual LAN"

_agronomy, doi:10.3390/agronomy12071565_

Round 1
Reviewer 1 Report
Although the article is interesting, the English must be completely revised, since the grammatical tense used is the past (use the present or the form "has been shown, or has shown". Plurals are forgotten, there are some missing spaces. In my opinion, there is redundant information that makes it difficult to read the article, so I ask that the graphs be simplified and supplemented with tables.
In the figures of the spatial distribution it is NOT possible to see what temperature each color is, which is essential to understand the distribution of temperature, likewise it should indicate if that day was cold, cloudy, hot, rainy or something relevant that happened . There should also be a photo of the place where the greenhouse is located to understand why the distribution of temperatures is so variable in space (Are there trees or mountains that provide shade?) The greenhouse is very small and I do not understand how it is that there is a distribution space so scattered.
In the interpolation that uses correlations there should be more details of how the correlation is taken, what is the separation factor between samples and how the non-deterministic part of the algorithm is treated. The interpolation gives ESTIMATES not Simulations.
There is missing information about the crop
The abstract must be corrected (see attached file) There are many corrections in the attached file that must be made. In the file the problems are highlighted or marked or commented

Author Response
Responses are placed behind the references in the article. Please see the attachment.

Reviewer 2 Report
This research is based on different monitoring procedures of greenhouses. In particular, temperature samples are interpolated by the Kriging algorithm. It is an adequate methodology but there is a lack of information about:
1. The requirements of data to employ Kiring algorithm like normal distribution and stationary of sources..
2. The way to adjust Kiring to the sampling methods must be defined Which model was selected to define the variance between odes? A spherical model? Which are the values of sill, nugget and range? How they were defined?
This information is needed to define clearly the methodology employed. After this different transmission methods (LAN or wifi) were compared and the sensibility of the obtained prediction with respect to the number of sampling sensors was analysed. It is an interesting procedure and in agreement with nowadays hot topics like IoT.
The paper format is good and the English style adequate. Just this improvement in the methodology is required to be an adequate paper for the journal.
Author Response

(The authors gave the same response as above.)

Round 2
Reviewer 2 Report
After different changes, the paper is adequate for publication.